# The opportunistic intracellular bacterial pathogen *Rhodococcus equi* elicits type I interferon by engaging cytosolic DNA sensing in macrophages

**Krystal J. Vail**[1,2], **Bibiana Petri da Silveira**[3], **Samantha L. Bell**[1], **Noah D. Cohen**[3], **Angela I. Bordin**[3], **Kristin L. Patrick**[1], **Robert O. Watson**[1] *

1 Department of Microbial Pathogenesis and Immunology, Texas A&M Health Science Center, Bryan, Texas, United States of America, 2 Department of Veterinary Pathology, Texas A&M University, College Station, Texas, United States of America, 3 Department of Large Animal Clinical Sciences, Texas A&M University, College Station, Texas, United States of America

* robert.watson@tamu.edu

**Data Availability Statement:** All relevant data are within the manuscript and its Supporting Information files.

## Abstract

*Rhodococcus equi* is a major cause of foal pneumonia and an opportunistic pathogen in immunocompromised humans. While alveolar macrophages constitute the primary replicative niche for *R. equi*, little is known about how intracellular *R. equi* is sensed by macrophages. Here, we discovered that in addition to previously characterized pro-inflammatory cytokines (e.g., *Tnfa*, *Il6*, *Il1b*), macrophages infected with *R. equi* induce a robust type I IFN response, including *Ifnb* and interferon-stimulated genes (ISGs), similar to the evolutionarily related pathogen, *Mycobacterium tuberculosis*. Follow up studies using a combination of mammalian and bacterial genetics demonstrated that induction of this type I IFN expression program is largely dependent on the cGAS/STING/TBK1 axis of the cytosolic DNA sensing pathway, suggesting that *R. equi* perturbs the phagosomal membrane and causes DNA release into the cytosol following phagocytosis. Consistent with this, we found that a population of ~12% of *R. equi* phagosomes recruits the galectin-3,-8 and -9 danger receptors. Interestingly, neither phagosomal damage nor induction of type I IFN require the *R. equi*'s virulence-associated plasmid. Importantly, *R. equi* infection of both mice and foals stimulates ISG expression, in organs (mice) and circulating monocytes (foals). By demonstrating that *R. equi* activates cytosolic DNA sensing in macrophages and elicits type I IFN responses in animal models, our work provides novel insights into how *R. equi* engages the innate immune system and furthers our understanding how this zoonotic pathogen causes inflammation and disease.

## Author summary

*Rhodococcus equi* is a facultative intracellular bacterial pathogen of horses and other domestic animals, as well as an opportunistic pathogen of humans. In human patients,

**Funding:** This work was supported by funds from the National Institutes of Health, grant 1R01AI125512 (ROW), the Texas A&M University T3 grant (ROW, KLP, AIB; https://t3.tamu.edu/), the Link Equine Research Endowment at Texas A&M University (NDC; https://vetmed.tamu.edu/), TAMU CVM Core Facility grant (KJV; https://vetmed.tamu.edu/), and NIH training grant 5T32OD011083-10 (KJV). The funders had no role in study design, data collection and analysis, decision to publish, or preparation of the manuscript.

**Competing interests:** The authors have declared that no competing interests exist.

*Rhodococcus* pneumonia bears some pathological similarities to pulmonary tuberculosis, and poses a risk for misdiagnosis. In horses, *R. equi* infection has a major detrimental impact on the equine breeding industry due to a lack of an efficacious vaccine and its ubiquitous distribution in soil. Given the prevalence of subclinical infection and high false positive rate in current screening methods, there exists a critical need to identify factors contributing to host susceptibility. Here, we use a combination of bacterial genetics and animal models to investigate innate immune responses during *R. equi* infection. We found that *R. equi* modulates host immune sensing to elicit a type I interferon response in a manner resembling that of *M. tuberculosis*. We also found that the danger sensors galectin-3, -8, and -9 are recruited to a population of *R. equi*-containing vacuoles, independent of expression of VapA. Our research identifies innate immune sensing events and immune transcriptional signatures that may lead to biomarkers for clinical disease, more accurate screening methods, and insight into susceptibility to infection.

## Introduction

*Rhodococcus equi* is a gram positive, intracellular bacterial pathogen that causes severe, potentially fatal respiratory disease in young horses (*Equus caballus*) less than 6 months of age. *R. equi* infection has a major detrimental impact on the equine breeding industry for several reasons: it is nearly ubiquitous in the soil of some facilities, there is not currently an efficacious vaccine, and early diagnosis is challenging[1]. Virtually all foals are exposed by inhalation of contaminated soil. While many develop subclinical infection, 18–50% of foals develop pneumonia but recover with treatment, and 2–5% perish [2–5]. Those that do not succumb to disease develop lifelong immunity and active infection is rare in adult horses [6].

*R. equi* is also a pathogen of humans, causing a pneumonia that radiographically and pathologically resembles pulmonary tuberculosis (TB), as well as extrapulmonary infections[7–9]. The majority of human cases manifest as pneumonia and occur in immunocompromised individuals, such as those with impaired cell-mediated immunity due to HIV infection [10] or immunosuppression therapy related to organ transplantation [11]. However, a growing number of cases have been reported in immunocompetent humans, less than half of which develop pulmonary lesions [8,9]. Over 50% of human infections are derived from porcine- or equine-adapted strains, indicating that most human *R. equi* infections are zoonotic [10,12].

Upon inhalation, *R. equi* survives and replicates within alveolar macrophages in a phagosomal compartment that fails to mature into a lysosome, resulting in the establishment of the *R. equi*-containing vacuole. Previous studies have shown that *R. equi* pathogenesis depends in large part on expression of bacterial virulence factors[13–16] and production of pro-inflammatory cytokines [17–19], but the nature of the innate immune milieu generated by *R. equi* infection remains ill-defined.

The toll-like receptor (TLR) family is a vital component of innate immunity against microbial pathogens. TLR signaling, through adaptor proteins such as MyD88 and TRIF, as well as transcription factors like NF-κB, is central to pro-inflammatory cytokine production[20]. Several lines of evidence demonstrate the importance of host expression of pro-inflammatory cytokines such as IFN-γ and TNF-α in *R. equi* pathogenesis [18]. While most laboratory strains of mice are resistant to *R. equi*, treatment with monoclonal antibodies against IFN-γ, TNF-α, or both fail to clear *R. equi* infection, develop pulmonary lesions, and succumb to disease [18,19,21]. Likewise, *R. equi* replication is reduced in equine monocyte derived macrophages primed with IFN-γ or TNF-α prior to infection [22]. The requirement for pro-inflammatory cytokine production in clearing *R. equi* was further illustrated by Darrah and colleagues, who

showed that IFN-γ deficient (*Ifng*$^{-/-}$) mice infected with low dose *R. equi* were hypersusceptible to infection (died 13 days post-infection), and mice with impaired nitric oxide (*Nos2*$^{-/-}$) or superoxide (*Gp91*$^{phox-/-}$) production were even more susceptible and died by days 7.5 and 9.5 post infection, respectively [23]. Together, these studies suggest that IFN-γ activates macrophages to produce reactive oxygen species that limit intracellular replication and kill *R. equi* [23].

While signaling via IFN-γ, a type II IFN, is important for macrophage activation and control of bacterial infection, type I IFN can act as a negative regulator of host defenses against intracellular bacterial infection[24]. This paradigm is particularly evident in mycobacterial infection, in which a type II IFN signature is associated with mild pathogenesis and increased pathogen clearance during both *Mycobacterium tuberculosis* and *Mycobacterium leprae* infection and a type I IFN response is correlated with diffuse lepromatous leprosy and active tuberculosis in humans [25,26]. Both *M. leprae* and *M. tuberculosis* replicate within a modified phagosome, which they permeabilize via their ESX-1 virulence-associated secretion systems, to interact with the host macrophage. It is increasingly appreciated that numerous intracellular bacterial pathogens, including *M. tuberculosis*, *L. monocytogenes*, and *F. tularensis*, activate cytosolic DNA sensing and induce type I IFN signaling through similar mechanisms [24]. Specifically, the cytosolic DNA sensor, cGAS is activated by binding bacterial DNA during infection with *M. tuberculosis* [27–30], *F. novicida*[31], *Legionella pneumophila* [32] and *Chlamydia* spp. [33]. Activation of cGAS leads to the production of the second messenger, cGAMP, a cyclic dinucleotide (CDN) that binds the adapter molecule STING. Other bacterial species such as *L. monocytogenes* [31,34,35], *M. tuberculosis* [30], and *C. trachomatis* [36] can also bypass cGAS and produce cyclic dinucleotides capable of directly activating STING, leading to type I IFN signaling.

Virulent *R. equi* possess an extrachromosomal virulence plasmid containing a *vap* pathogenicity island[37]. *R. equi* lacking this plasmid are unable to replicate within macrophages and do not cause disease in foals [38,39]. The *vap* pathogenicity island encodes the Vap family of proteins, the best characterized of which is VapA, which is required, but not sufficient to confer virulence in foals[40]. Additionally, *R. equi* encodes a type VII (commonly known as ESX) secretion system[37], but whether this system is active and how it contributes to the macrophage response to *R. equi* is not known.

In spite of the detrimental impact this important pathogen has on the equine breeding industry, data on macrophage sensing of *R. equi* is incomplete, and to date, studies of this pathogen have centered on extracellular macrophage receptors. Here, we sought to investigate whether this vacuolar pathogen triggers innate immune responses in the macrophage cytosol. Additionally, we sought to characterize the transcriptional response triggered by *R. equi* during *ex vivo* infection in murine macrophages as well as *in vivo* in mice and in foals. Using RNA-seq,we found that numerous type I IFN genes are upregulated during *R. equi* infection, and that this bacterium induces phagosomal permeabilization in a way that recruits galectin danger sensors. While both galectin recruitment and the type I IFN gene expression profile was not dependent on expression of the *R. equi* virulence-associated protein A (VapA), type I IFN production did require the cGAS/STING/TBK1 signaling axis. Furthermore, we found that a type I IFN program was induced *in vivo* in both a mouse model as well as in an equine infection model. These data provide evidence that *R. equi* activates the cytosolic DNA sensing pathway during macrophage infection and suggest that type I IFN signaling may be critical for *R. equi* pathogenesis.

## Results

### *R. equi* induces pro-inflammatory cytokine expression during macrophage infection

Proinflammatory cytokines, particularly TNF-α and IFN-γ have been shown to be important in controlling *R. equi* infection in foals. To begin to define the nature of the macrophage innate immune response to *R. equi* infection, we set out to compare innate immune responses in primary macrophages derived from both foals and mice to those elicited in RAW 264.7 cells, a genetically tractable macrophage-like cell line in which our lab has developed tools to knock-down or -out key components of the innate immune system [27,41–44]. In past studies using RAW 264.7 cells with Mtb, *Salmonella*, and *Listeria*, we have found that activation of both TLR and cytosolic nucleic acid sensing pathways induce transcriptional responses comparable to primary macrophages [27,41,43–45]. To determine whether *R. equi* infection followed a similar paradigm, we first compared pro-inflammatory cytokine induction at the transcript level in both primary murine bone marrow derived macrophages (BMDMs) and equine bronchoalveolar cells (consisting of 85% macrophages) (S1A Fig). Primary cells were infected with virulent *R. equi* (ATCC 330701+) at a multiplicity of infection (MOI) of 5. Total RNA from uninfected and infected cells was harvested after 4 and 8h (which we have extensively characterized as key early induction timepoints downstream of innate immune sensing events in multiple intracellular bacterial infection models) [27,28,46]) and RT-qPCR was performed to measure expression of innate immune genes at different time points post-infection. At both 4- and 8h, *R. equi*-infected primary murine macrophages had robust induction of the pro-inflammatory cytokines *Tnfa*, *Il1b* and *Il6*, with *Tnfa* and *Il1b/Il6* peaking at 4h and 8h, respectively, consistent with activation of pattern recognition receptors such as TLR2 (Fig 1A). Equine BAL cells infected with *R. equi* also showed similar robust induction of *Tnf* at 4h and 8h (Fig 1B), albeit with slightly delayed kinetics. We next performed the same experiment in the RAW 264.7 murine macrophage cell line and found that although there was slightly less robust induction of *Tnfa*, *Il1b*, and *Il6*, the transcripts were induced with similar timing when compared to the equine and murine primary cells (Fig 1C). We also tracked NF-κB activation by measuring phosphorylation of NF-κB in RAW 264.7 cell lysates by immunoblot analysis and observed NF-κB phosphorylation at 2-, 4- and 6h with peak phosphorylation at 4h post-infection (Fig 1D). Based on these data, we concluded that RAW 264.7 macrophages respond to *R. equi* similarly to primary bone marrow derived and equine alveolar macrophages and chose to employ RAW 264.7 cells as a powerful genetic model to investigate the innate immune response to *R. equi*. To these ends, we infected *Myd88* shRNA knockdown and scramble shRNA (SCR) control RAW 264.7 cells with *R. equi* and found that expression of these cytokines was dependent, at least in part, on MyD88 (*Myd88* KD, 65% efficiency) (Fig 1E) as the *Myd88* knockdown cells had significantly reduced levels of *Il1b*, *Il6* and *Tnfa* compared to SCR controls (Fig 1F). These findings are consistent with a study by Darrah and colleagues, who also reported TLR2- and MyD88-dependent production of pro-inflammatory cytokines during RAW 264.7 macrophage infection with *R. equi* [17].

### Transcriptomics uncovers upregulation of type I IFN in *R. equi*-infected murine macrophages

Having validated RAW 264.7 cells as a model to understand the *R. equi*-macrophage interface, we next sought to more unbiasedly examine the macrophage innate immune response to *R. equi* infection. To this end, we turned to RNA-seq to assess global gene expression changes following infection. High-throughput RNA sequencing was carried out on 3 biological replicates

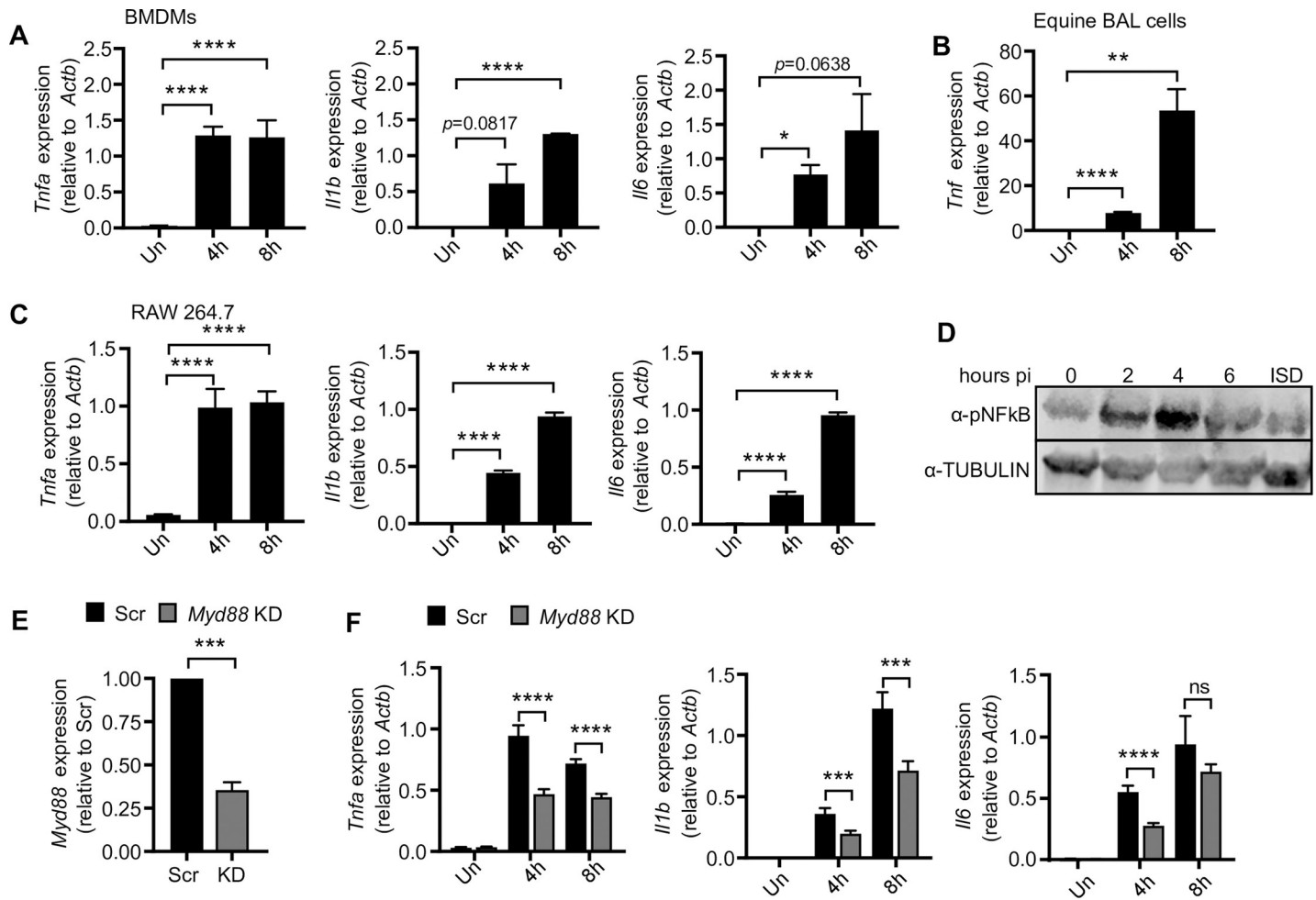

**Fig 1. *R. equi* induces pro-inflammatory cytokine expression during macrophage infection.** (A) RT-qPCR of *Tnfa*, *Il1b* and *Il6* in murine BMDMs at indicated time post *R. equi* infection. (B) RT-qPCR of *Tnf* in equine bronchoalveolar cells (BAL) at indicated time post-infection with *R. equi*. (C) As in (A) but in RAW 264.7 macrophages. (D) Immunoblot of pNFκB in *R-equi*-infected or ISD-transfected macrophages, with TUBULIN as a loading control. (E) RT-qPCR of *Myd88* expression in RAW 264.7 *Myd88* KD cells relative to Scr control. (F) RT-qPCR of *Tnfa*, *Il1b* or *Il6* in *R. equi*-infected *Myd88* KD macrophages. All RT-qPCRs are the mean of 3 replicates ± SD, n = 3 and are representative of at least 2 independent experiments. Statistical significance was determined using Students' t-test. *p < 0.05, **p < 0.01, ***p < 0.001, ****p < 0.0001, ns = not significant.

of uninfected and *R. equi*-infected macrophages (S1 Table). After filtering transcripts with a fold change of > ± 2 (*p*< 0.05), we identified 390 genes that were upregulated and 65 genes that were downregulated in *R. equi*-infected macrophages compared to uninfected controls (Fig 2A). Interestingly, we noticed that this expression profile had considerable overlap with that of macrophages infected with Mtb (4h post-infection), with 167 *R. equi*-induced genes also upregulated in Mtb-infected macrophages, and 6 genes downregulated in both groups (Fig 2B). Consistent with previous findings ([17,47,48] and Fig 1), we observed significant upregulation of numerous canonical pro-inflammatory cytokines (*Il1a*, *Il1b*, and *Tnf*), chemokines (*Cxcl2*, *Ccl4*, *Ccl3*, *Cxcl10*), inflammasome genes (*Nlrp3*), and prostaglandins (*Ptgs2*) at 4h post-*R. equi* infection (Fig 2C).We also observed upregulation of several antiviral genes that are induced downstream of the IRF (interferon regulatory factor) family of transcription factors (*Rsad2*, *Ifit1*).To validate RNA-seq gene expression changes during *R. equi* infection, representative upregulated (*Lif*, *Nlrp3*) and downregulated (*Mafb* and *S1pr1*) transcripts were measured by RT-qPCR (Fig 2D). Because some innate immune transcripts can peak after 4h,

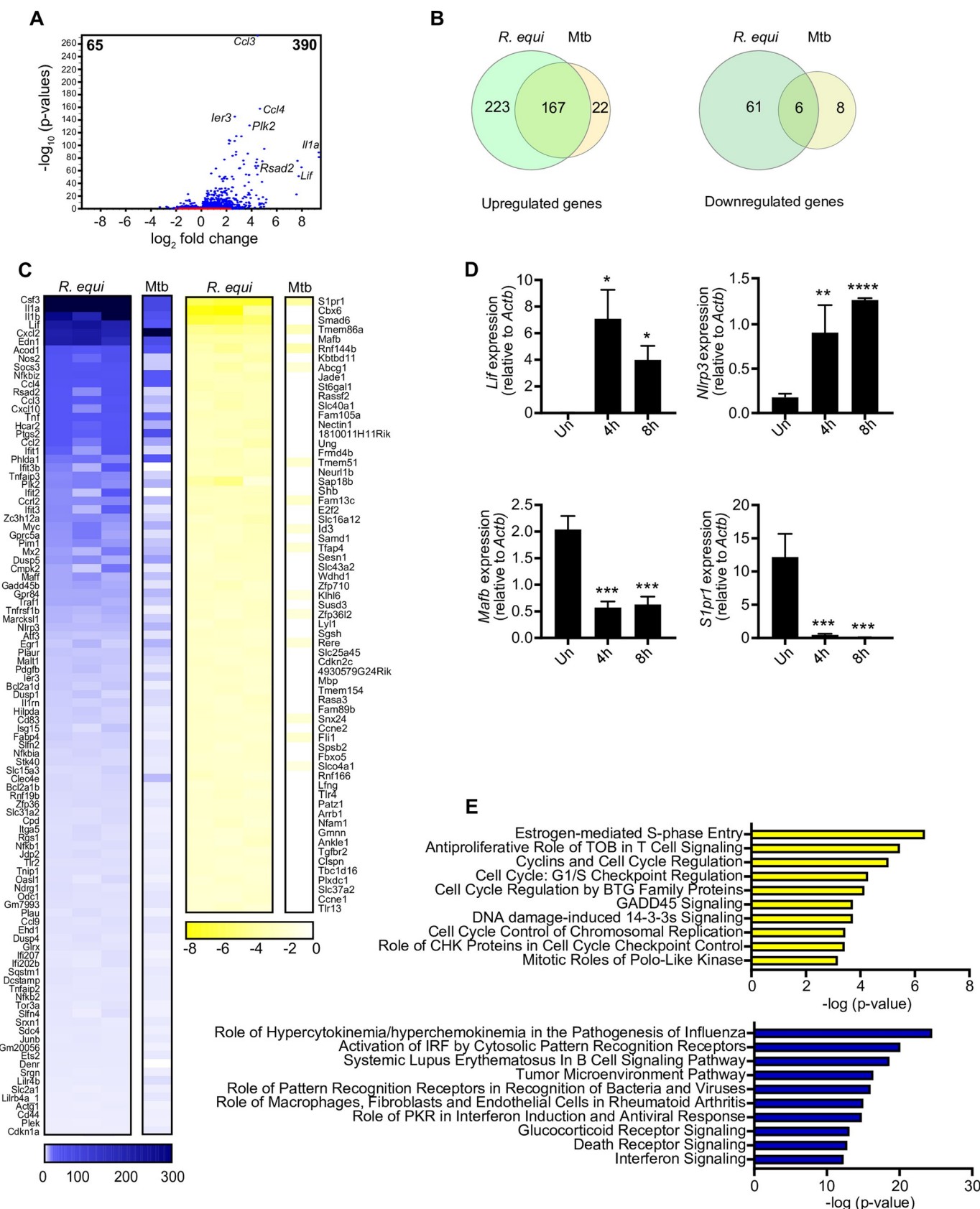

**Fig 2. RNA-seq analysis reveals upregulation of pro-inflammatory cytokines and type I interferon response genes in *R. equi*-infected murine macrophages.** (A) Volcano plot showing gene expression analysis of RAW 264.7 macrophages infected with R. equi. x axis shows fold change of gene expression and y axis shows statistical significance. Downregulated genes are plotted on the left and upregulated genes are on the right. (B) Venn diagram comparing differentially expressed genes in RAW 264.7 genes during *R. equi* and Mtb infection. Upregulated genes are shown in the left Venn diagram and downregulated genes are shown in the right. (C) Heatmap showing gene expression analysis of RAW 264.7 macrophages infected for 4h with *R. equi* or Mtb compared to uninfected cells. Each column for *R. equi* represents a biological replicate. Mtb is shown as average of 3 replicates. Blue genes are upregulated in infected cells, yellow genes are downregulated in infected cells. (D) RT-qPCR validation of upregulated genes (*Lif, Nlrp3*) and downregulated genes (*Mafb, S1pr1*) in RAW 264.7 macrophages at 4 and 8h post infection with *R. equi*. (E) Ingenuity pathway analysis of gene expression changes in RAW 264.7 macrophages infected with *R. equi*. Downregulated genes are shown on top in yellow and upregulated genes are shown on the bottom in blue. (A-E) represent 3 biological replicates ± SD, n = 3. For RT-qPCRs, statistical significance was determined using Students' t-test. *p < 0.05, **p < 0.01, ***p < 0.001, ****p < 0.0001, ns = not significant. For DGE statistical significance was determined using the EDGE test in CLC Genomics Workbench. Significant differentially expressed genes were those with a p <0.05 and fold change of <±2.

we measured gene expression at both 4- and 8h following infection; overall, these RT-qPCR results were consistent with the RNA-seq data.

We next asked what pathways are enriched for differentially expressed genes in uninfected vs. *R. equi*-infected macrophages. Unbiased canonical pathway analysis (Ingenuity Pathway Analysis (IPA, Qiagen)) of the biological processes most enriched during *R. equi* infection showed strong upregulation of genes related to innate immune signaling ("Role of pattern recognition receptors in recognition of bacteria and viruses", "Interferon signaling", "Glucocorticoid receptor signaling"), cell death ("Death receptor signaling"), and tumor pathogenesis ("Tumor microenvironment pathway") (Fig 2E). Pathways enriched for downregulated genes were primarily related to cell cycle regulation ("Estrogen-mediated S phase entry", "Cyclins and cell cycle regulation", "Cell cycle G1/S checkpoint regulation", "Cell cycle regulation by BTG family proteins", "Cell cycle control of chromosomal replication") (Fig 2E). Intriguingly, viral pathogenesis-related pathways, specifically "Role of hypercytokinemia/hyperchemokinemia in the pathogenesis of Influenza" and "Role of PKR in IFN induction and antiviral response," were among the most enriched pathways in our IPA analysis of upregulated genes. Together, these findings began to suggest that antiviral type I interferon (IFN) expression, in addition to pro-inflammatory cytokines and chemokines, may play an important role in *R. equi* pathogenesis.

To further investigate the type I IFN response induced by *R. equi*, we first assessed the dynamics of *Ifnb* expression over a time-course of *R. equi* macrophage infection in murine BMDMs and in equine BAL cells by RT-qPCR. We observed strong induction of *Ifnb* in both cell types, which peaked at 4h in BMDMs (Fig 3A) but showed highest induction at 8 hours in equine BAL cells (Fig 3B).

*R. equi*'s ability to survive and replicate within macrophages is largely dependent on a ~ 90 kb virulence plasmid and the <u>v</u>irulence <u>a</u>ssociated <u>p</u>roteins (Vaps) it encodes. The best characterized, VapA, is required for virulence in foals and promotes intracellular survival [13,15], as plasmid-cured strains of *R. equi* fail to replicate inside macrophages and do not cause inflammatory cell death [14]. To investigate if VapA is required for inducing type I IFNs in response to *R. equi* infection, we infected RAW 264.7 macrophages with isogenic strains of *R. equi* either with (33701+) or without (33701-) VapA at a MOI of 5 for 4- and 8h and measured gene expression. We validated the expression or absence of VapA using RT-qPCR with primers directed against a segment of the VAPA gene (Fig 3C and S2A Fig). We observed similar *Ifnb* expression in macrophages infected with *R. equi* 33701- and macrophages infected with virulent *R. equi* 33701+ (Fig 3D), suggesting that VapA does not drive type I IFN expression in macrophages.Similarly, *Isg15* expression in macrophages infected with *R. equi* 33701- was virtually identical to macrophages infected with virulent *R. equi* 33701+ (Fig 3E). We also measured *Tnfa, Il1b, Il6* and *Isg15* expression and found that macrophages infected with plasmid-cured *R. equi* (33701-) had reduced expression of *Tnfa* at 8h, and of *Il6* at both 4- and 8h after

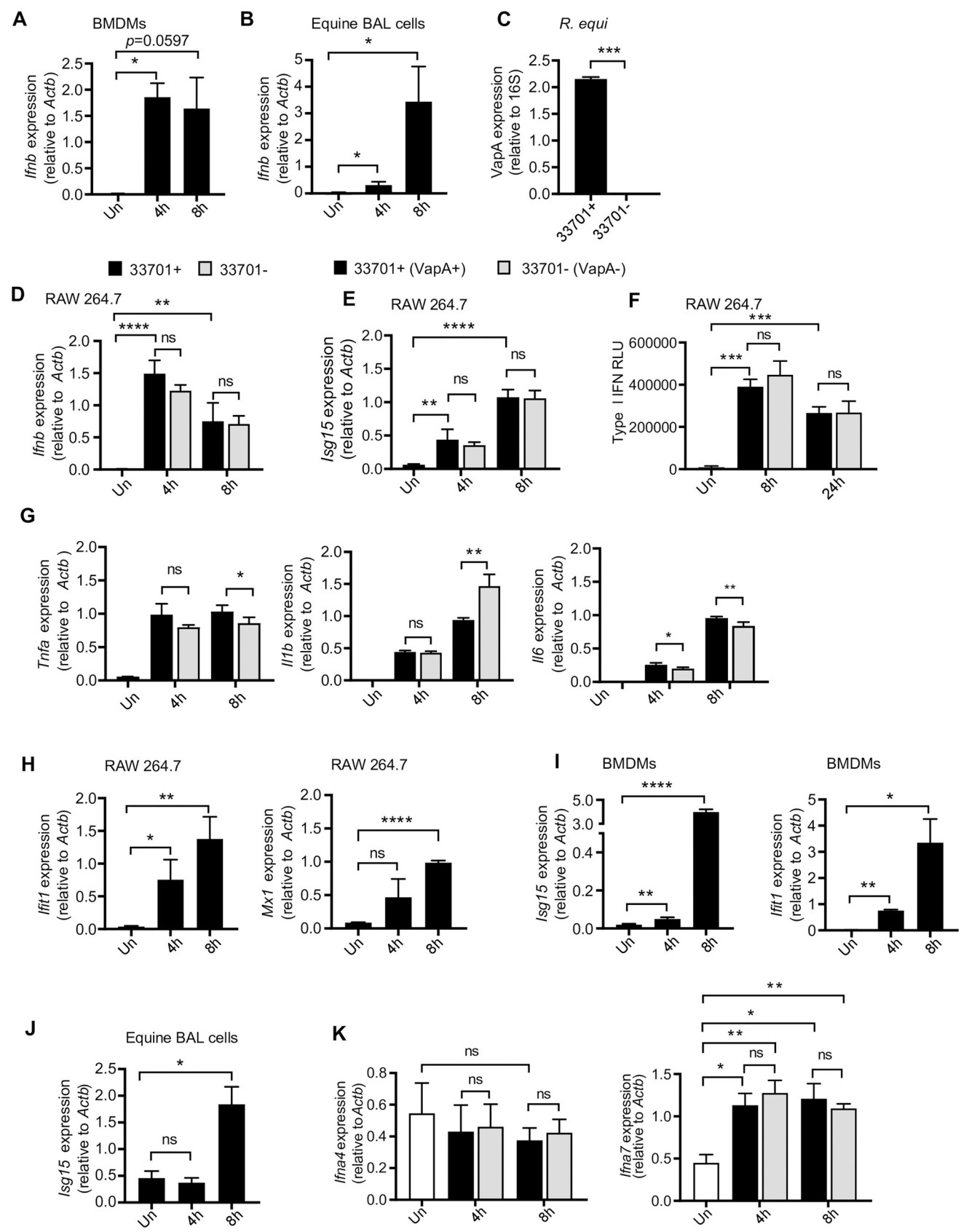

**Fig 3. *R. equi* induces type I IFN expression during macrophage infection.** (A) RT-qPCR of *Ifnb* in *R. equi*-infected murine BMDMs. (B) As in (A) but in equine bronchoalveolar lavage cells. (C) RT-qPCR of VapA in *R. equi* 33701+ and 33701-. (D) RT-qPCR of *Ifnb* in RAW 264.7 cells infected with *R. equi* with (33701+) or without (33701-) VapA. (E) As in (D) but of *Isg15*. (F) ISRE reporter cell assay with relative light units (RLU) measured as a readout for secreted type I IFN protein in *R. equi*-infected macrophages. (G) As in (A) but *Tnfa*, *Il1b*, and *Il6*. (H) RT-qPCR of *Ifit1* and *Mx1* in RAW 264.7 cells infected with *R. equi* 33701+. (I) As in (A) but *Isg15* and *Ifit1*. (J) As in (B) but *Isg15*. (K) As in (D) but *Ifna4* and *Ifna7*. All RT-qPCRs are representative of at least 2 independent experiments and are the mean of 3 replicates ± SD, n = 3. Statistical significance was determined using Students' t-test. $^{*}$p < 0.05, $^{**}$p < 0.01, $^{***}$p < 0.001, $^{****}$p < 0.0001, ns = not significant.

infection (Fig 3G). Consistent with a previous study using *R. equi* strain ATCC 103 in mouse peritoneal macrophages [48], macrophages infected with *R. equi* 33701- had greater *Il1b* expression at 8h post-infection (Fig 3G).

To determine whether transcriptional upregulation of *Ifnb* leads to elevated protein levels in *R. equi* infected cells, we used an IFN-stimulated response element (ISRE) luciferase reporter cell line as a readout for secreted IFN-α/β from *R. equi*-infected RAW 264.7 macrophages. Consistent with the transcriptional changes described above (Fig 3A, 3B and 3D), we observed robust production of IFN-α/β protein in response to *R. equi* infection. As expected, secretion of type I IFN was not dependent on *R. equi* expression of VapA (Fig 3F).

Secreted IFN-β is recognized in an autocrine and paracrine manner through the IFN-β receptor (IFNAR1/2) and results in downstream expression of interferon stimulated genes (ISGs). Thus, we next measured expression of several ISGs, including *Isg15*, *Ifit1* and *Mx1*, over the same time course in BMDMs in addition to RAW 264.7 macrophages and observed significant induction of these genes 4h post-infection with higher induction at 8h in both cell types (Fig 3H and 3I). Additionally, we observed induction of *Isg15* at 8h post infection in equine BAL cells (Fig 3J). Therefore, we concluded that *R. equi* infection elicits a robust type I IFN response in macrophages.

Although *Ifnb* is our primary readout for type I IFN gene expression and the major type I IFN mediator expressed in macrophages, it is also possible that other interferons, such as *Ifna* could also be induced by *R. equi*, contributing to the type I IFN response, and if so, could be dependent upon VapA expression. To test this, we measured expression of *Ifna4* and *Ifna7* in *R. equi*-infected RAW 264.7 macrophages. No significant induction of *Ifna4* was observed at 4- or 8h following infection (Fig 3K). Induction of *Ifna7* was not dependent upon *R. equi* expression of VapA (Fig 3K). These findings demonstrate that while VapA may influence pro-inflammatory cytokine expression, it is not required for induction of type I IFNs during *R. equi* infection of macrophages.

## TBK1 is required for type I IFN induction in response to *R. equi* infection in primary murine macrophages

*Ifnb* is induced in a number of ways, including downstream of extracellular (TLR4 sensing of LPS (which activates the TRIF-TBK1 axis)), endosomal (e.g., TLR9 sensing of CpG DNA, TLR3 sensing of double-stranded RNA), or cytosolic (e.g., cGAS sensing of double-stranded DNA) sensing pathways, each of which trigger phosphorylation and activation of the IRF family of proteins, primarily IRF3 and IRF7. IRF7 is expressed at low levels in macrophages until its expression is induced downstream of type I IFN, while IRF3 is expressed and activated downstream of pathogen associated molecular patterns [49,50]. To determine if IRF3 is activated in response to *R. equi* infection, we infected RAW 264.7 cells (MOI of 50) and measured phosphorylated IRF3 (Ser396) by immunoblot. We detected IRF3 phosphorylation at 2-, 4- and 6h, post-*R. equi* infection, peaking at 4h (Fig 4A). We also examined STAT1 activation, which occurs downstream of IFNAR engagement, and found strong phosphorylation at 4-, and 6h post-infection, with peak activity at 4h (Fig 4A).

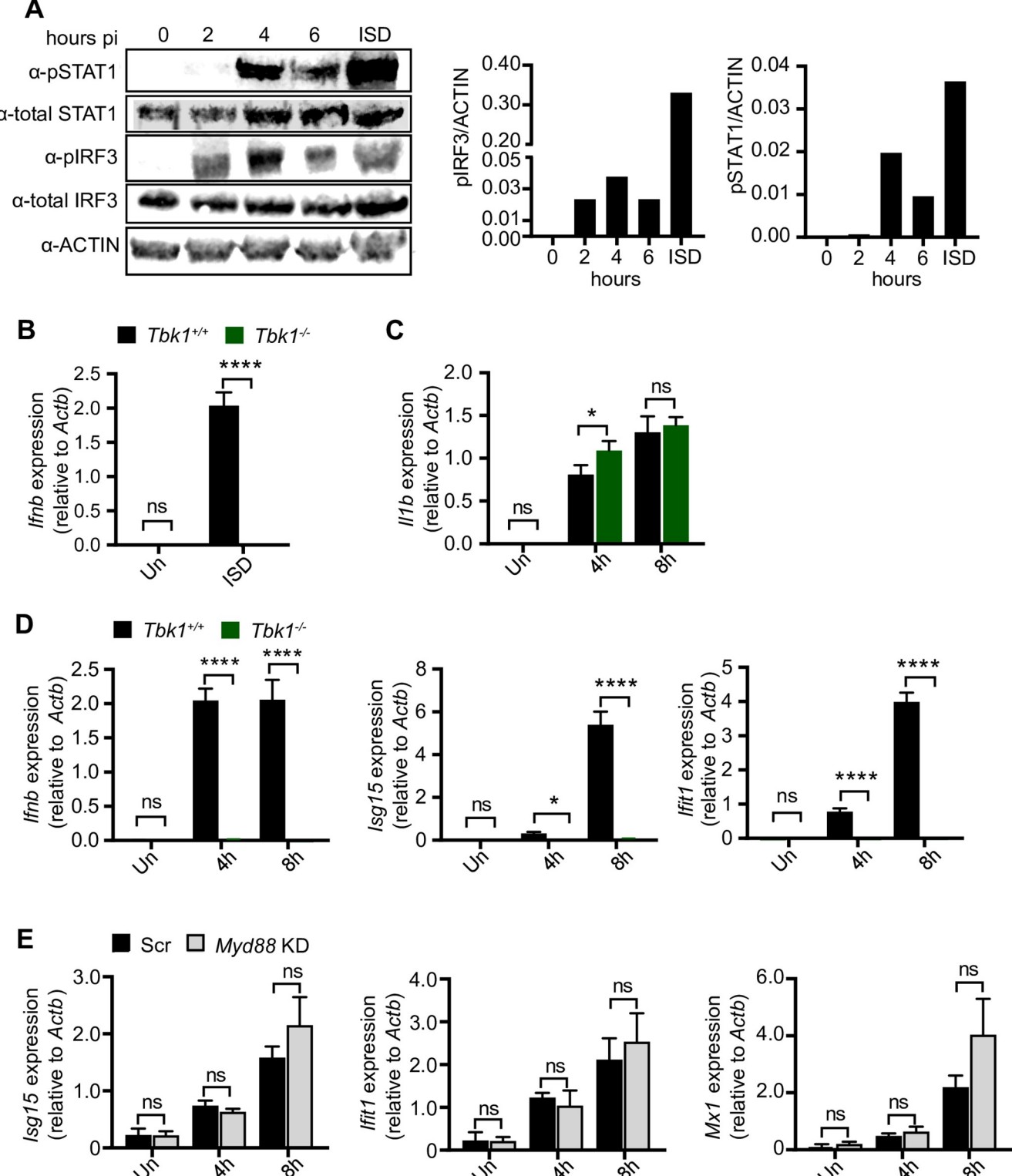

**Fig 4. TBK1 is required for type I IFN induction in response to *R. equi* infection in primary murine macrophages.** (A) Immunoblot of pSTAT1, total STAT1, pIRF3 and total IRF3 in RAW 264.7 macrophages infected or not (0) with *R. equi* for 2, 4 or 6h or transfected with ISD for 4h. ACTIN was used as a loading control. (B) RT-qPCR of *Ifnb* in control (*Tbk1*^+/+^*Tnfr*^-/-^) and KO (*Tbk1*^-/-^*Tnfr*^-/-^) BMDMs transfected with ISD for 4h. (C) RT-qPCR of *Il1b* in TBK1 BMDMs uninfected or infected with *R. equi* for the indicated times. (D) As in (C) but *Ifnb*, *Isg15* and *Ifit1*. (E) RT-qPCR of *Isg15*, *Ifit1* and *Mx1* in *Myd88* KD macrophages infected with *R. equi* for the indicated times. All RT-qPCRs are representative of at least 2 independent experiments and are the

mean of 3 replicates ± SD, n = 3. Immunoblots are representative of at least 3 independent experiments. Statistical significance was determined using Students' t-test. *p < 0.05, **p < 0.01, ***p < 0.001, ****p < 0.0001, n.s. = not significant.

Having observed activation of IRF3 in *R. equi*-infected macrophages (Fig 4A), we hypothesized that the upstream innate immune kinase TBK1 is required for type I IFN production. To test the contribution of TBK1 in type I IFN signaling during *R. equi* infection, we isolated BMDMs from mice lacking the kinase TBK1. Because *Tbk1* deletion in C57BL/6 mice causes a TNF receptor-dependent embryonic lethality, we used *Tbk1⁻/⁻/Tnf⁻/⁻* double KO (TBK1 KO) mice to assess the contribution of TBK1 and compared them to *Tbk1⁺/⁺/Tnf⁻/⁻* controls [27,51]. As expected, *Tbk1⁻/⁻* macrophages failed to induce *Ifnb* in response to transfection with the double-stranded DNA agonist ISD [32] (Fig 4B). While *R. equi*-infected *Tbk1⁻/⁻* BMDMs had no difference in expression of the pro-inflammatory cytokine *Il1b* compared to controls (Fig 4C), they showed almost no induction of *Ifnb*, *Isg15* and *Ifit1* at 4- and 8h after infection (Fig 4D), indicating that TBK1 is required for induction of a type I IFN response following infection with *R. equi*.

Because autocrine sensing of IFN-β by IFNAR elicits ISG expression via the STAT1/STAT2 transcription factors, we predicted that loss of IFNAR would abrogate the type I IFN response in *R. equi* infected macrophages. Compared to WT controls, the *Ifnar1⁻/⁻* macrophages had no differences in pro-inflammatory cytokine levels after *R. equi* infection (S3B Fig). However, *R. equi*-infected *Ifnar1⁻/⁻* macrophages displayed a modest reduction in *Ifnb* levels (S3C Fig), and ISG expression (*Isg15*) was completely abrogated (S3C Fig). As a positive control, when we transfected *Ifnar1⁻/⁻* macrophages with ISD (which stimulates *Ifnb* production), there was minimal ISG induction (S3D Fig) consistent with IFNAR being required for ISG expression during *R. equi* infection.

We next investigated specific adapter proteins required for type I IFN induction during *R. equi* infection. Type I IFN production downstream of TLRs is mediated via MyD88/TRIF, while cytosolic DNA and RNA sensing signals through cGAS/STING and MAVS/RIG-I, respectively. MyD88 activates type I IFN downstream of TLR9 in response to unmethylated CpG DNA, while TRIF mediated activation of type I IFN occurs via TLR3 in response to dsRNA [20]. To begin to implicate one or more of these adapters in the *R. equi* type I IFN response, we first measured ISG expression (i.e., *Isg15*, *Ifit1* and *Mx1*) in *Myd88-* and *Trif-* knockdown macrophages (*Trif* KD, 70% efficiency) (Figs 4E and S3F) and detected no major differences in ISG expression in either TLR adapter knockdown cell line, with the exception of a slight but significant reduction in *Isg15* expression in *Trif* knockdown RAW 264.7 cells (S3G Fig). These results suggest that neither TLR9 nor TLR3 are the major drivers of type I IFN signaling during *R. equi* infection. Collectively, these results demonstrate that *R. equi* induces type I IFN through TBK1 and IFNAR, but that this occurs via a sensor other than TLR3 or 9.

## Cytosolic DNA sensing via cGAS/STING/TBK1 is required to induce type I IFN in macrophages infected with *R. equi*

Having ruled out type I IFN induction by several adapter proteins downstream of TLR sensors, we next asked whether *R. equi* infection directly stimulates cytosolic nucleic acid sensing. STING is an adaptor protein in the cytosolic DNA sensing axis that is activated by host cyclic dinucleotides like cGAMP produced by activated cGAS or by bacterial cyclic dinucleotides like c-di-AMP or c-di-GMP [24,52,53]. To investigate whether STING is required for *R. equi* induction of type I IFNs, we infected CRISPR-Cas9-generated STING knock out (KO) RAW 264.7 macrophages with *R. equi* and measured cytokine expression by RT-qPCR. In response

to *R. equi* infection, while STING KO macrophages induced *Tnfa* to comparable levels as control macrophages (expressing a GFP-targeting guide RNA), they failed to induce *Ifnb*, *Isg15*, or *Ifit1* at 4- or 8h post-infection (Fig 5A). As a positive control we transfected STING KO RAW 264.7 macrophages with ISD (to stimulate STING via cGAS) and found that these cells fail to induce *Ifnb* (S4A Fig) [54]. Using ISRE reporter cells, we also observed a significant reduction in the protein levels of IFN-α/β in the supernatants of STING KO macrophages compared to controls (S4B Fig). These data indicate that STING is absolutely required for type I IFN expression in response to *R. equi* infection in macrophages.

Some intracellular bacterial pathogens such as *Listeria monocytogenes* activate cytosolic sensing by producing cyclic dinucleotides (c-di-AMP) that bind to and activate the adaptor protein STING, which in turn activates TBK1 [55]. To determine if *R. equi* might activate the cytosolic DNA sensing pathway in a similar way, we searched the Kegg database and found that *R. equi* encodes a cyclic dinucleotide diadenylate cyclase that could potentially produce c-di-AMP [56]. To test whether *R. equi* stimulates STING directly via production and secretion of c-di-AMP or via activation of cGAS, which produces the cyclic dinucleotide cGAMP in response to binding cytosolic double-stranded DNA [33,57], we generated cGAS KO RAW 264.7 cells using CRISPR-Cas9. As with loss of STING, cGAS KO macrophages fail to induce *Ifnb* in response to ISD transfection (S4C Fig). Upon infection with *R. equi*, cGAS KO macrophages had reduced levels of *Ifnb*, *Isg15* and *Ifit1* transcripts(Fig 5B), and a significant but less dramatic reduction in *Tnfa* (Fig 5B). To test the contribution of cGAS to the type I IFN response in human macrophages, we used shRNA knockdown of cGAS and STING in the human monocyte cell line U937 [27], with at least 30% KD efficiency (Figs 5C and S4C). As with RAW 264.7 KO cells, STING and cGAS KD U937 cells showed a significant reduction in type I IFN 8 hours after infection (Figs 5D and S4D). At the protein level, cGAS KO macrophages had defective IFN-α/β production as measured by ISRE reporter cells, but notably, production was not completely ablated (S4F Fig). However, when we measured IFN-β protein levels at 0-, 8- and 24h post-infection by ELISA, we observed a complete loss of IFN-β in supernatants from both cGAS and STING KO macrophages (Fig 5E). To ensure this phenotype was not a consequence of variations in bacterial burden, as well as to investigate the role of cytosolic DNA sensing in cell-intrinsic control of *R. equi* replication, we measured CFUs in *R. equi*-infected cGAS and STING KO macrophages at 1, 12, and 24 hours. We found that loss of type I IFN production in cGAS and STING KO macrophages was not due to impaired bacterial engulfment or replication as bacterial burden was similar in control, STING KO and cGAS KO macrophages over the course of infection (S4G Fig).

We also measured STAT1 activation in *R. equi*-infected control, STING KO and cGAS KO macrophages by immunoblot, probing for phosphorylated (activated) STAT1 (Tyr701). We found that in control macrophages STAT1 was robustly phosphorylated at 4- and 6h post-infection, but phospho-STAT1 was undetectable in STING KO or cGAS KO macrophages at all time points (Fig 5F). Together, these results demonstrate that *R. equi* infection induces a type I IFN response through activation of the cGAS/STING/TBK1 axis.

## Vacuolar acidification and *de novo* bacterial protein synthesis are important for activation of the type I IFN response during *R. equi* infection

The cGAS/STING/TBK1 axis can be activated by a number of mechanisms, including perturbation and release of host mitochondrial DNA (mtDNA) or release of bacterial DNA into the cytosol [58]. To begin to test whether *R. equi*-mediated cytosolic mtDNA leakage could be responsible for eliciting type I IFN production, we treated RAW 264.7 cells with the mtDNA synthesis inhibitor, 2'3' dideoxycytidine (ddC), which depletes mtDNA from cells over several

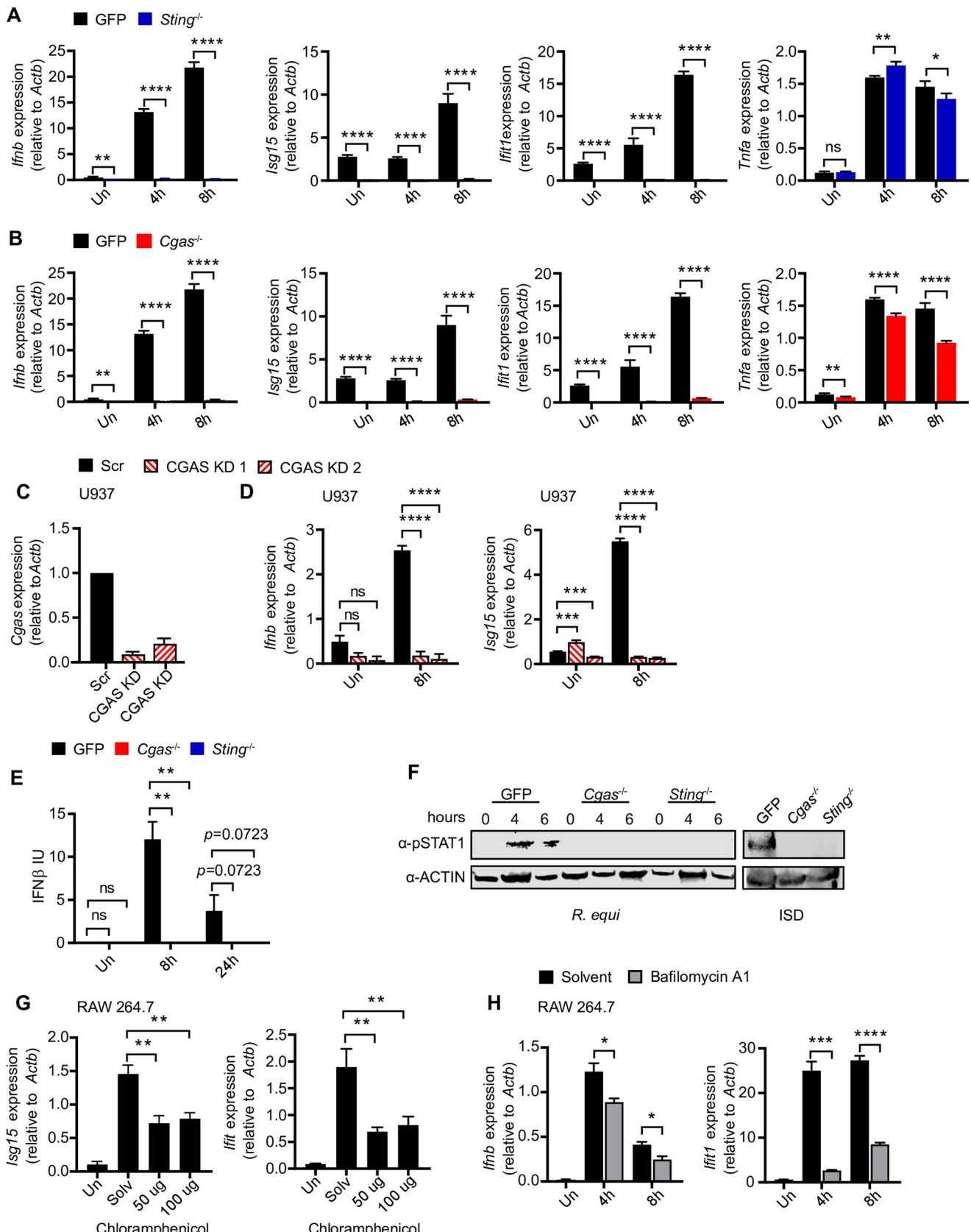

**Fig 5. The cytosolic DNA sensing axis of cGAS/STING/TBK1 is required to induce type I IFN during *R. equi* infection of macrophages.** (A) RT-qPCR of *Ifnb*, *Isg15*, *Ifit1* and *Tnfa* in GFP gRNA control or STING KO RAW 264.7 macrophages infected with *R. equi* for the indicated times. (B) As in (A) but in GFP gRNA control and cGAS KO RAW 264.7 macrophages. (C) RT-qPCR of *Cgas* in cGAS KD U937 cells compared to Scr control. (D) As in (E) RT-qPCR of *Ifnb* and *Isg15* in *R. equi* infected cGAS KD U937 cells. (E) IFN-β protein ELISA of RAW 264.7 macrophages infected with *R. equi* for the indicated times. (F) Immunoblot of pSTAT1 in GFP gRNA control, cGAS KO and STING KO RAW 264.7 cells infected *R. equi* for the indicated times or transfected with ISD for 4h. ACTIN was used as a loading control. (G) RT-qPCR of *Isg15* and *Ifit1* in RAW 264.7 cells 8h post infection with *R. equi* treated with the indicated concentration of chloramphenicol. (H) RT-qPCR of *Ifnb* and *Ifit1* in *R. equi*-infected RAW 264.7 cells treated with 100nM bafilomycin A. All RT-qPCRs are representative of at least 2 independent experiments and are the mean of 3 replicates ± SD, n = 3. Immunoblots are representative of at least 3 independent experiments. Statistical significance was determined using Students' t-test. $^*p < 0.05$, $^{**}p < 0.01$, $^{***}p < 0.001$, $^{****}p < 0.0001$, n.s. = not significant.

passages [44]. We observed no reduction in type I IFN expression in *R. equi*-infected macrophages treated with ddC compared to untreated macrophages (S4I Fig), indicating that mtDNA is likely not responsible for activation of the cGAS/STING/TBK1 axis during *R. equi* infection, at least at these early time points.

We next considered that macrophage internalization of *R. equi* might trigger synthesis of bacterial proteins and/or gene products involved in induction of type I IFN by cytosolic DNA sensors. To test whether *de novo* protein synthesis is required for *R. equi*-induced type I IFN expression, we treated *R. equi* cultures with chloramphenicol, an antibiotic that inhibits *de novo* bacterial synthesis. Interestingly, we observed a reduction in *Ifnb*, *Isg15* and *Ifit1* in the presence of chloramphenicol compared to those treated only with solvent (Figs 5G and S4L), indicating that *R. equi de novo* protein synthesis contributes to the type I IFN response in macrophages. These results are also consistent with the hypothesis that *R. equi* has evolved specific adaptations to interface with the cytosol to engage the cytosolic DNA sensing pathway.

One common cue that induces bacterial virulence gene expression programs is a lower vacuolar pH [59–61]. To determine whether acidification of the *R. equi*-containing vacuole is required to induce the type I IFN gene expression program, we blocked vacuolar acidification by treating RAW 264.7 macrophages with the vacuolar $H^+$-ATPase inhibitor bafilomycin A1. We found bafilomycin A1 inhibited *Ifnb* expression at both 4h and 8h (Fig 5H). Bafilomycin A1-treated cells also showed reduced expression of the ISG, *Ifit1* at 8h (Fig 5H). These results show that acidification of the *R. equi*-containing vacuole is required to induce a robust type I IFN response and further suggest that a lower vacuolar pH might be required to fully induce expression of bacterial virulence factors that are responsible for engaging with the cytosol of the infected macrophage.

## Galectin-3,-8, and -9 are recruited to *R. equi*-containing vacuoles

Because *R. equi* is thought to be confined by a phagosomal membrane until 24h post-infection in macrophages[62], it was puzzling to observe activation of the cytosolic DNA sensor cGAS early during infection that also required bacterial *de novo* protein synthesis. Previous studies discovered that Mtb, another "vacuolar" pathogen, permeabilizes its phagosomal membrane using its ESX-1 secretion system [28]. Therefore, we hypothesized that *R. equi* may also permeabilize its phagosomal membrane to allow for communication with the host cytosol, resulting in activation of cytosolic DNA sensors. We recently reported that the cytosolic glycan-binding proteins galectin-3, -8, and -9 access the lumen of damaged Mtb-containing vacuoles following ESX-1-mediated permeabilization [42]. To investigate whether *R. equi* phagosomal membranes are sufficiently damaged during infection to recruit these galectins, we infected RAW 246.7 cells stably expressing 3xFLAG-tagged galectins [42]. Specifically, we used galectins-3, -8, -9 because they have been shown to colocalize with Mtb, *L. monocytogenes*, *Salmonella enterica* serovar Typhimurium, and *Shigella flexneri* [42,63]; *R. equi*-containing vacuole was used as a negative control. Each of these galectin cell lines were infected with virulent GFP-expressing *R. equi*(MOI 5) and over a time

course of 4-, 8-, and 16h, cells were fixed and imaged by immunofluorescence microscopy [64]. We found that galectin-3, -8, and -9 but not galectin-1 were all recruited to *R. equi* to varying degrees. Recruitment of galectins-8 and -9 peaked at 8h post-infection, with galectins-8 and -9 recruited to ~10% and ~6% of *R. equi*, respectively (Fig 6A and 6B). Galectin-3 recruitment peaked at 4h post-infection, with recruitment to ~12% of *R. equi* at 4h and declining to ~5% by 16h. Curiously, by 4h post infection, galectin-3, -8, and -9 formed puncta in the cytosol of *R. equi* infected cells (Fig 6C), although the nature of these puncta is unclear. Galectin-1 formed rare puncta but did not associate with *R. equi*(Fig 6C).Our detection of a galectin-positive population of *R. equi* suggests that phagosomal membrane damage and access to the cytosol occurs as early as 4h following infection. Interestingly, at 16h, many galectin-positive *R. equi* had markedly reduced GFP expression compared to galectin-negative bacteria, suggesting they had potentially been killed or lysed (S5 Fig).

Given that the *R. equi* virulence factor VapA has been shown to permeabilize lysosomal membranes to modulate lysosome pH [13] and is required for survival and replication within macrophages [14], we hypothesized that VapA might be required for permeabilizing the phagosomal membrane and permitting galectin recruitment. To test this, we infected 3xFLAG tagged galectin-8 RAW246.7 cells with GFP-expressing, plasmid-cured *R. equi* 103-, which does not express VapA[65]. The plasmid cured strain of *R. equi* had similar recruitment of galectin-8 as VapA expressing *R. equi* (Fig 6D), indicating that the *R. equi* phagosomal membrane permeabilization does not require VapA and occurs via an alternative mechanism. Taken together, these data indicate that the *R. equi*-containing vacuole is permeable and accessible to the macrophage cytosol, which could enable detection of bacterial-derived ligands by cGAS and other cytosolic sensors.

## *R. equi* induces both a pro-inflammatory and type I IFN expression program *in vivo*

Having shown that *R. equi* induces type I IFNs in macrophages, we next sought to determine how the type I IFN response contributes to *R. equi* pathogenesis *in vivo*. Because we can exert a degree of control over the murine environment, we began by infecting 8-week-old C57BL/6 mice with *R. equi* by intraperitoneal (IP) injection. At 5d post-infection, we harvested lungs, mesenteric lymph nodes, spleens, and peritoneal cells. Age-matched mice injected with an equal volume of PBS (Un) served as negative controls. Bacteria were recovered from the spleen, mesenteric lymph nodes, and lungs, indicating that *R. equi* disseminated from the point of initial infection (Fig 7A). We measured spleen weights as a readout for inflammatory responses and observed a ~1.5-fold increase in splenic weight in infected mice compared to those injected with PBS (Fig 7B). We also measured pro-inflammatory cytokines and type I IFNs in the spleen and peritoneal cells isolated from *R. equi*-infected mice 5d post-infection by RT-qPCR. Consistent with our observations in macrophages, spleens and peritoneal cells from mice infected with *R. equi* had significantly elevated levels of both pro-inflammatory cytokines and ISGs compared to uninfected controls (Fig 7C).

Because mice are relatively resistant to *R. equi* infection, and since horses are the primary natural host and most physiologically relevant model, we next turned to an equine model of infection. We collected pre-infection, baseline blood samples from 28-day-old horses and assessed their lungs for pulmonary lesions by thoracic ultrasonography. We next infected foals intrabronchially via endoscopy [66–68] with 1 x 10$^6$virulent *R. equi* and monitored them for 3 weeks; age-matched uninfected foals served as negative controls. At 21d post-infection, we again collected blood and evaluated lungs by thoracic ultrasonography to monitor disease progression or resolution. To investigate if *R. equi* induces a type I IFN program in horses, we

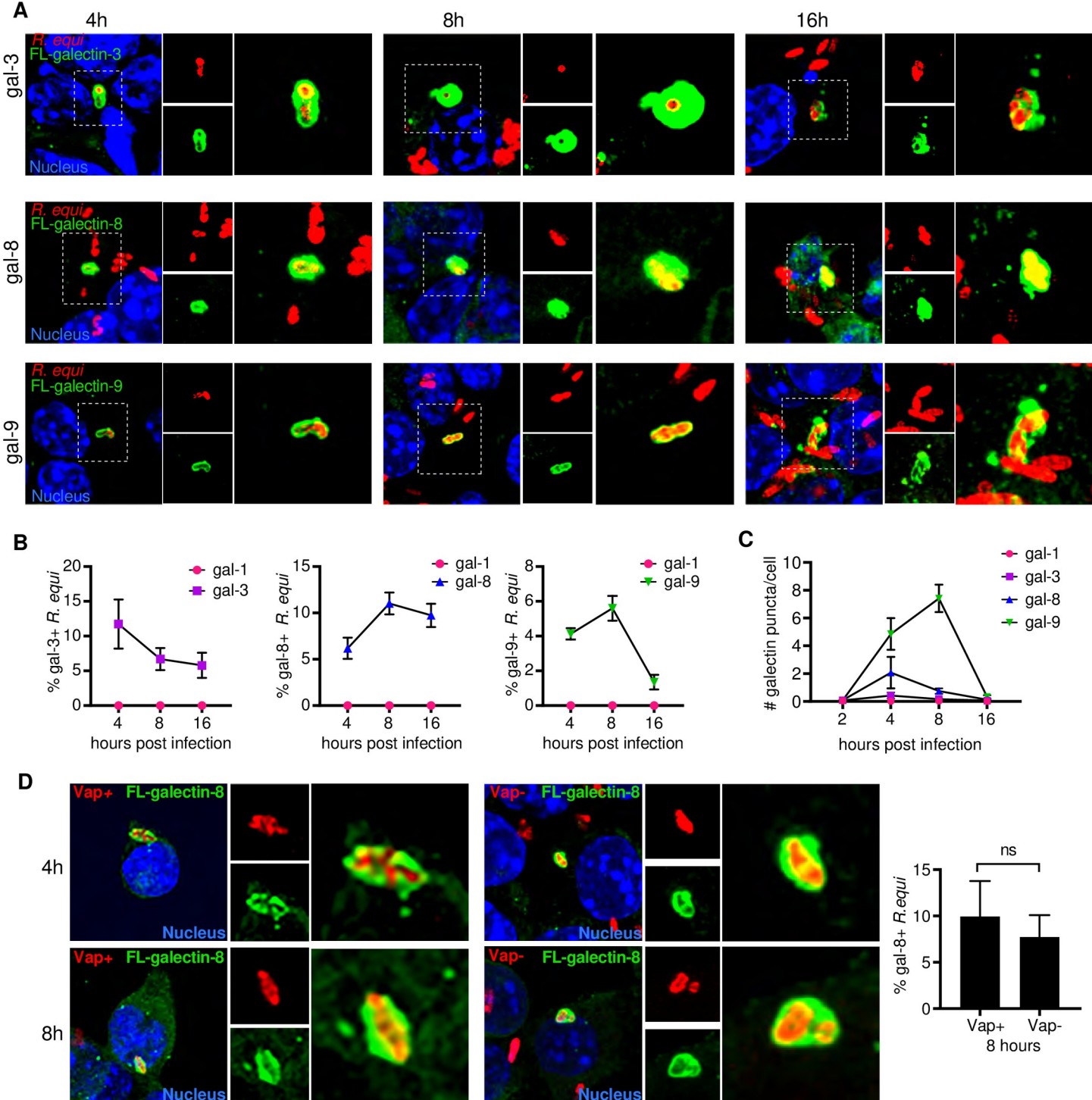

**Fig 6. Galectins are recruited to *R. equi*.** (A) Immunofluorescence (IF) of RAW 264.7 cells stably expressing 3XFLAG (FL)-tagged galectin-3, -8, or -9 infected with GFP expressing *R. equi* 103+ (MOI-5) for the indicated times. (B) Quantification of galectin-1, -3, -8, or -9 recruitment to *R. equi* at the indicated times. (C) Quantification of galectin-1, -3, -8, or -9 puncta in *R. equi*-infected RAW 264.7 cells at the indicated times. A minimum of 100 cells were quantified per coverslip. (D) IF of RAW 264.7 cells stably expressing 3X FL-tagged galectin-8 infected with GFP expressing *R. equi* 103+ or 103- (MOI-5) for the indicated time and quantification of galectin-8 positive *R. equi* in at 8h expressed as percent of total *R. equi*. IF images are representative of 3 independent experiments. Quantification is the percent of positive bacteria with at least 100 bacteria quantified per coverslip. Error bars are ±SEM. Statistical significance was determined using Students' t-test. *p < 0.05, n.s. = not significant.

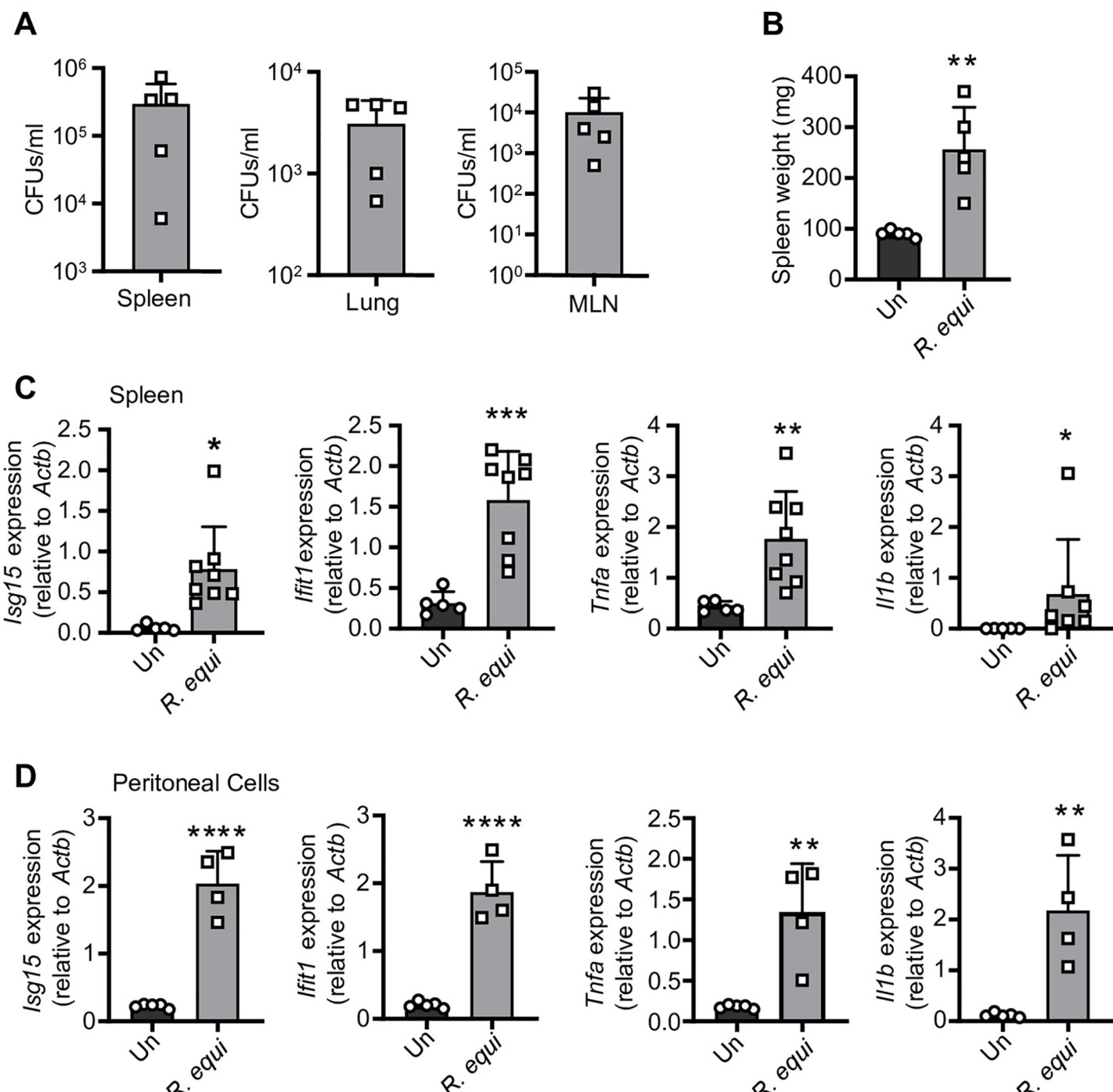

**Fig 7. *R. equi* induces both a pro-inflammatory and type I IFN expression program in a mouse model of infection.** (A) CFUs/gram of the spleen, lung and mesenteric lymph nodes of mice 5d post-infection. Each square represents an individual mouse. (B) Spleen weight in mg of mice 5d post *R. equi* infection. (C) RT-qPCR of *Isg15*, *Ifit1*, *Tnfa* and *Il1b* in the spleen of mice uninfected or infected with *R. equi*. (D) As in (C) but peritoneal cells. Mouse RT-qPCRs represent 5 biological replicates ± SD. For all experiments in this study, statistical significance was determined using Mann-Whitney test. $^*p < 0.05$, $^{**}p < 0.01$, $^{***}p < 0.001$, $^{****}p < 0.0001$, ns = not significant.

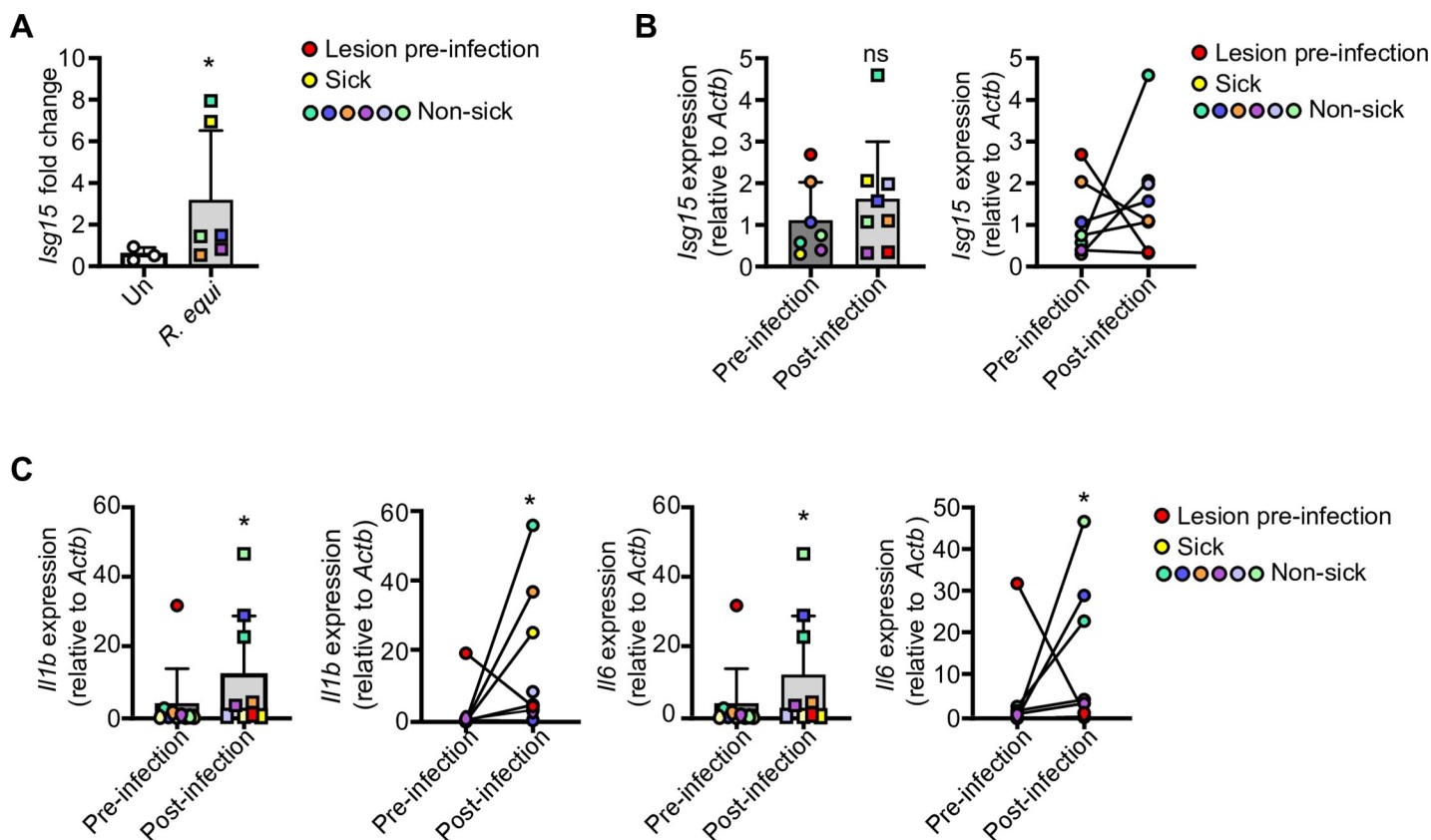

**Fig 8. *R. equi* induces type I IFN response in equine monocytes.** (A) RT-qPCR of *Isg15* in sham infected and *R. equi* infected foals at 21d post-infection normalized to baseline *Isg15* levels. Each dot represents an individual foal. (B) As in (A) but comparing baseline *Isg15* levels to 21d post-infection in infected foals. Red dot indicates foal with lesions prior to experimental infection and yellow dot indicates foal that became clinically ill after infection. (C) As in (B), but for *Il1b* and *Il6*. Horse RT-qPCRs represent >3 biological replicates ± SD, uninfected n = 3, infected n = 6. For all experiments in this study, statistical significance was determined using Mann Whitney test. *p < 0.05, **p < 0.01, n.s. = not significant.

isolated monocytes from peripheral blood samples taken at 0- and 21d post-infection and measured ISGs by RT-qPCR. To rule out age-related changes in ISG levels, we first compared uninfected and *R. equi*-infected foals at 21d post-challenge, normalizing individual post-challenge values to baseline levels. Uninfected foals had no significant induction of ISGs between the baseline and 21d post-challenge samples (Fig 8A). However, we observed a significant induction of *ISG15* in infected foals, with infected foals falling into subgroups with modest induction or high induction. Notably, the sole foal that became clinically ill following experimental challenge fell within the high induction subgroup (denoted by yellow square in Fig 8A). Since individual foals varied in their response to infection and having shown that pre- and post-challenge cytokine levels were similar in uninfected foals, we next focused on infected foals and compared baseline *ISG15* levels with post-infection levels. We observed an upward trend in *ISG15* induction compared to pre-infection samples, but it did not reach statistical significance (Fig 8B). We also observed a robust induction of pro-inflammatory cytokines (*IL1B*, *IL6*) in *R. equi*-infected foals (Fig 8C). Given that *R. equi* is ubiquitous in the environment and likely shed by mares, and given that 2 foals exhibited evidence of pulmonary lesions prior to experimental infection (denoted by red circles in Fig 8B and 8C), it is possible that foals were exposed to environmental *R. equi*, which may have contributed to innate immune activation during the 0d blood collection.Nonetheless, these results indicate that *in vivo R. equi* infection results in activation of the type I IFN response in both mouse and equine models of infection.

## Discussion

Because innate immune responses dictate adaptive immune outcomes, defining the innate immune milieu generated in response to bacterial pathogens like *R. equi* is a crucial step to understanding how it causes disease. Here, using an unbiased approach we show that in addition to triggering a pro-inflammatory cytokine response, *R. equi* elicits type I IFN signaling in macrophages during the acute stages of infection. We show that intracellular *R. equi* triggers the cytosolic DNA sensing pathway in a cGAS/STING/TBK1-dependent manner. Activation of the cytosolic DNA sensing pathway following *R. equi* infection leads to phosphorylation of IRF3 and initiation of a type I IFN response within 4h of infection, while intracellular bacteria are still replicating within the *R. equi*-containing vacuole [16,69]. These findings support a model whereby *R. equi* persists in a modified vacuole capable of interacting with the macrophage cytosol and engaging DNA sensors and other innate immune proteins.

Our observation that vacuolar *R. equi* induces type I IFN expression in a VapA-independent manner via activation of the DNA sensing pathway parallels that of Mtb in macrophages. Mtb uses its type VII secretion system, ESX-1, to permeabilize its vacuolar membrane, permitting liberation of extracellular mycobacterial DNA into the macrophage cytosol where it triggers cGAS [27]. Genetic evidence indicates that like Mtb, *R. equi* also encodes a type VII secretion system, although the function of this system in *R. equi* remains to be determined[56,70]. Further studies focused on investigating the function of *R. equi*'s ESX secretion systems will be important for uncovering the mechanism of cytosolic DNA sensing during *R. equi* macrophage infection. Mtb ESX-1 effectors ESAT-6 and CFP-10, together with the mycobacterial lipid PDIM, are thought to form pores in the phagosome after their secretion [71,72]; while the presence or function of orthologous effectors in *R. equi* might suggest a similar mechanism, *R. equi* could also have additional, unrelated ways of accessing the cytosol to communicate with and manipulate host macrophages. Using readouts and phenotypes we identified here, genetic screens could be designed to elucidate these mechanisms and uncover novel *R. equi* effectors and effector functions.

The recruitment of the glycan binding proteins galectin-3, -8, and -9 to a population of *R. equi* indicates that the *R. equi*-containing vacuole is damaged to the extent that luminal glycans are exposed to and recognized by cytosolic danger sensors as early as 4h post infection. This is consistent with Mtb recruitment of these galectins, which occurs as early as 3h post-infection [42], while *Salmonella enterica* serovar Typhimurium recruits galectin-3, -8, and -9 to broken vacuoles as early as 1h post-infection [63]. Interestingly, we observed an accumulation of galectin puncta that are unassociated with *R. equi* bacteria. It is possible that virulence factors responsible for permeabilizing *R. equi*'s phagosome are secreted and act in trans to also damage endosomal compartments independent of the *R. equi*-containing vacuole.

Our finding that cGAS is required for induction of type I IFN signaling during *R. equi* infection indicates that DNA is the major contributor to this activation in macrophages; however; the source of this DNA remains to be determined. Given the data supporting damaged *R. equi* phagosomes, it is likely that bacterial DNA is released into the cytosol. Recent evidence indicates that Mtb infection induces release of host mitochondrial DNA into the cytosol, contributing to the type I IFN response during infection [73,74]. However, this is unlikely during *R. equi* infection, at least at the time points we tested, since we observed no reduction in type I IFN production upon mtDNA depletion. While some bacteria can directly activate STING through secretion of c-di-nucleotides, and while *R. equi* does seem to encode deadenylatecyclases, our data suggest that this is not the mechanism of type I IFN induction under the conditions or time points we examined. Because cGAS KOs do not induce type I IFNs, it suggests that without host-derived cGAMP, no bacterial CDNs are present to activate STING and induce type I IFNs. Likewise, this effectively rules out cytosolic RNA sensing at these time points. However, our experiments do not directly rule out the

contribution of mitochondrial DNA, bacterial derived CDNs, or RNA-sensing at later time points during infection. Regardless of the mechanism, our findings add *R. equi* to a growing list of intracellular bacterial pathogens that elicit type I IFN through cGAS (Mtb [27,29,75], *Chlamydia* spp. [33], *F. novicida*[31]) and/or STING (*L. monocytogenes*[76], Mtb [30], and *Chlamydia trachomatis* [36]), suggesting there is a selective advantage for bacteria to elicit this host response, likely as a way to block IL-1 activity via ILRα [77], an ISG and/or limit IFN-γ responses [25].

A key virulence determinant of *R. equi* is the conjugative virulence plasmid which hosts a 21-kb pathogenicity island encoding Vaps. One Vap, VapA, promotes *R. equi* survival in macrophages by inhibiting phagosomal maturation [65], inducing lysosome biogenesis [15], and contributing to lysosome membrane permeabilization [13]. Interestingly, we found that disruption of the phagosomal membrane, activation of cytosolic DNA sensing, and production of type I IFNs was not dependent on expression of Vap A. This apparent discrepancy may be explained by the transient nature or small size of VapA-induced membrane leaks, with VapA-induced membrane lesions ranging from 0.37 nm at pH 6.5 to 1.05 nm at pH 4.5, which are only large enough for the passage of ions [13]. Further studies investigating other bacterial mutants may help elucidate additional bacterial factors underlying in the host-pathogen interactions of *R. equi* and macrophages.

One potential outcome of TBK1 activation during *R. equi* infection is autophagic targeting [27,46,78]. Autophagy, and selective autophagy in particular, functions as an antimicrobial mechanism that promotes lysosomal degradation of intracellular bacterial pathogens [79]. Although the recruitment of galectins and activation of TBK1 suggests that a population of *R. equi* could potentially be targeted and destroyed by this mechanism, we found that bacterial replication and survival were unaltered by loss of cGAS or STING.

Another outstanding question is how induction of type I IFN signaling influences *R. equi* pathogenesis. It will be interesting to determine the transcriptional signature in foals with severe disease to test the hypothesis that foals that succumb to infection have a robust, or even hyperinduced, type I IFN signature. While we observed induction of type I IFN in response to *R. equi* infection during our equine experiments, we were unable to draw clear conclusions regarding connections between type I IFN levels and disease outcome largely due to the small study size. All animals in the equine study ultimately cleared infection without treatment, and only one foal became clinically ill following experimental challenge; however, this animal did have highly induced ISGs. Future studies in both mouse and horse models specifically centered on the balance between type I and type II IFN will be key in understanding the clinical implications of a type I IFN-skewed response, and whether such a signature correlates with reduced IFN-γ, impaired control of bacterial replication, or detrimental disease outcomes. Identifying factors associated with negative *R. equi* disease outcomes will help determine ways to expedite diagnoses, promote positive disease outcomes, and reduce patient mortality.

## Methods

### Ethics statement

All animal protocols used for this study were reviewed and approved by the Texas A&M University Institutional Animal Care and Use Committee. Mice were euthanized by $CO_2$ asphyxiation and cervical dislocation according to the 2020 AVMA Guidelines for the Euthanasia of Animals. Equine studies were nonterminal.

### Animals

All experiments for this study were reviewed and approved by the Texas A&M University Institutional Animal Care and Use Committee (AUP# 2019–0083). Mice were kept on a 12h

light/dark cycle and provided food and water *ad libitum*. Mice were group housed (maximum 5 per cage) by sex on ventilated racks in temperature-controlled rooms. TBK1 KO mice (*Tnfr1*$^{-/-}$ and *Tbk1*$^{-/-}$/*Tnfr1*$^{-/-}$) [80] were generously provided by the Akira lab. *Ifnar* KO mice (B6(Cg)-Ifnar1Ifnar1tm1.2Ees/J) stock #028288) were purchased from The Jackson Laboratories (Bar Harbor, ME). Foals and their mares were housed individually in stalls and separately from other mare and foal pairs for 1 week following experimental infection. After 1 week, these mare/foal pairs were transferred back to their original pasture.

## Cell culture

Bone marrow-derived macrophages (BMDMs) were differentiated from bone marrow cells isolated by washing mouse femurs with 10 ml DMEM. Harvested cells were centrifuged for 5 min at 1000 rpm and resuspended in BMDM media ((DMEM, 20% FBS, 1 mM Sodium pyruvate, 10% MCSF conditioned media). Cells were counted and plated at $5 \times 10^6$ in 15 cm non-TC treated dishes in 30 ml complete media and fed with an additional 15 ml of media on Day 3. On Day 7, cells were harvested with 1x PBS-EDTA. RAW 264.7 cells were purchased from ATCC and the cell line was minimally passaged in our laboratory to maintain genomic integrity, and new cell lines generated from these low passage stocks. Cell lines were passaged no more than 10 times and tested negative for mycoplasma contamination. Cells were cultured in complete media (DMEM, 10% FBS, 2% HEPES buffer). cGAS and STING KD U937 cells have been previously described[27]. U937 (ATCC CRL-1593.2) cells were cultured in suspension in complete media (RMPI with L-glutamine, 10% FBS, 20 mM HEPES). For selection, 250 ug/ml hygromycin was added to the media. Before infection, U937 cells were plated on dishes and cultured with 100 ng/ml phorbol-12-myristate 13-acetate (PMA) (Sigma) for at least 48 hours to induce differentiation.

## Broncho-alveolar lavage

The BAL procedure and processing of BAL fluid was performed as previously described [66]. Briefly, a BAL catheter (Jorgensen Labs, Colorado) was passaged down the trachea and lodged into a bronchus and the air cuff was inflated. Two 60-ml syringes containing 120 ml of sterile saline solution (0.9% NaCl) were infused through the catheter and immediately aspirated, then the process was repeated 60-ml at a time until a total of 360 ml of saline was infused. A 3-ml aliquot of the BAL fluid containing cells was separated for differential cytology by the Texas Veterinary Medical Diagnostic Laboratory (College Station, TX). BAL cells were centrifuged at 400 ×g for 10 min, washed once with PBS, and then re-suspended in DMEM containing 10% heat-inactivated horse serum (Gibco, Gaithersburg, MD), 1% l-glutamine–penicillin–streptomycin solution (PSG; Sigma Aldrich, St. Louis, MO) and amphotericin B (25 μg/ml; United States Pharmacopeia, Rockville, MD). Cells were then counted and diluted to a concentration of $1 \times 10^6$ live cells/ml.

## Monocyte isolation

Thirty ml of heparinized blood was incubated at room temperature for 30 min. The plasma layer containing white blood cells was removed, diluted with the same volume of PBS, and layered over Ficoll-Paque Plus (GE Healthcare, Uppsala, Sweden) for density gradient separation of peripheral blood mononuclear cells (PBMCs). PBMCs were washed 3 times with PBS, counted in an automated cell counter (Cellometer Auto T4, NexelomBioscience, Lawrence, MA), and suspended at a concentration of $3 \times 10^6$ cells/mL in RPMI-1640 (BioWhittaker, Lonza, Walkersville, MD, USA) containing 15% heat-inactivated horse serum, 1% Glutamax (Life Technology Corporation, Grand Island, NY, USA), 1% NEAA mixture (BioWhittaker,

Lonza, Walkersville, MD, USA), penicillin G (100 U/mL), and streptomycin (80 μg/mL). PBMCs were incubated at 37˚C and 5% $CO_2$ for 3h in a T75 flask, and then non-adherent cells were removed with warm PBS. Adherent cells were detached from the flask with Accutase (Innovative Cell Technology) for 10 min at room temperature and PBS. Cells were pelleted, washed with PBS to remove Accutase, and the adherent cells (monocytes) counted using the automated cell counter, pelleted again, and stored in Trizol (Invitrogen) at -80˚C until RNA extraction.

## Stimulations

RAW 264.7 cells and murine BMDMs were plated in 12 well dishes at $5x10^5$ cells/well and allowed to adhere overnight. Cells were stimulated with 100 ng/ml LPS or transfected with 1 μg/ml ISD for 4h.

## Bacterial strains

Unless otherwise noted, virulent *Rhodococcus equi* 33701+ (VapA +; ATCC reference strain; Rockville, MD) was used as the wild type strain. The plasmid cured (33701-) strain was previously described [81]. We validated the expression or absence of VapA using RT-qPCR with primers directed against a segment of the VAPA gene (VAPAF 5'-GTAGGATCC-GACCGTTCTTGATTCCGGTAG-3' and VAPAR 5'-CTAGTCGACCCGGAAGTACGTG-CAG-3'). *R. equi* ATCC 103 expressing GFP [64,65] were generously provided by Dr. Mary Hondalus.

## Macrophage infections

One colony forming unit (CFU) of *R. equi* was inoculated into 5 ml of brain-heart infusion (BHI) broth and shaken overnight at 37˚ C, then 0.5 ml of the culture was subcultured into 5 ml of fresh BHI overnight, shaken at 37˚C. The bacterial suspension was centrifuged at 1000 rpm for 10 minutes at 25˚C. The suspension was discarded and the pellets washed twice with 1 ml of phosphate buffered saline (PBS). The supernatant was discarded, the bacteria re-suspended in sterile PBS, and the concentration of bacteria determined spectrophotometrically at an optical density of 600 nm (OD 600) where OD 1.0 represents approximately $2x10^8$ CFU/ml. Concentrations were verified by plating the inoculum and counting CFUs. Bacterial suspensions were diluted to the desired concentration. Strains were confirmed to be virulent (VapA positive) by PCR before infection. Macrophages were plated in 12-well dishes at $5x10^5$ cells/well and allowed to adhere overnight. Macrophages were infected with virulent *R. equi* at a MOI of 5 for gene expression studies, MOI of 1 for CFU experiments or MOI of 50 for immunoblot analysis. Following infection, macrophages were centrifuged for 10 minutes at 1000 rpm, then incubated for 30 minutes at 37˚C. Noninfected macrophage monolayers were cultured under the same conditions. Media containing *R. equi* was removed and each well washed twice with PBS, then replaced with complete media and cultured at 37˚C until the appropriate time point. At each time point, supernatants were removed and monolayers were washed with PBS.

For chloramphenicol treatments, the concentration required to inhibit *R. equi* multiplication was obtained by treating cultures in BHI with serial dilutions of water-soluble chloramphenicol (Sigma C3175) ranging from 3.125μg to 100 μg and measuring the OD600 over time. 100 and 50 μg were selected because growth remained static at 100 μg. For infections, subcultures were grown until log phase, and following washing, the appropriate concentration of chloramphenicol was added to each inoculum and incubated at 37˚C for 30 minutes prior to

inoculation. Infections were carried out as above, with the appropriate concentration of chloramphenicol included in the media until harvest.

For bafilomycin A experiments, RAW 264.7 cells were treated with 100 nM bafilomycin A (InvivoGen) for 30 minutes prior to inoculation with *R. equi*. Reagents were maintained in media until harvest.

## Experimental infection of mice

Ten 8- to 12-week-old C57Bl/6 mice were transferred to an ABSL-2 holding facility and allowed to acclimate for one week prior to infection. For *R. equi* infections, at the time of infection, mice were injected intra-peritoneally with 100μl of *R. equi* suspension. Following infection, animals were monitored and weighed daily until the experimental end point 5d post-infection to detect physical signs of disease. Lungs, spleens, mesenteric lymph nodes and peritoneal cells were harvested aseptically. Samples for CFU experiments were placed in pre-weighed, 10 ml conical tubes containing 2 ml of PBS. Tubes were then weighed to obtain organ weight, and organs were homogenized. Serial dilutions of the homogenate were placed onto LB agar plates and allowed to grow for 24-48h at 37˚C to determine the bacterial concentration per gram of organ. Samples for gene expression studies were placed in 500 μl of Trizol reagent, homogenized and stored at -80˚C until further processing.

## Experimental infection of foals

For transendoscopic infection[66–68], foals were sedated using intravenous (IV) injection of xylazine hydrochloride (0.5 mg/kg; Vedco, St. Joseph, MO) and IV butorphanol tartrate (0.02 mg/kg; Zoetis, Florham Park, New Jersey). An aseptically-prepared video-endoscope with outer diameter of 9 mm was inserted via the nares into the trachea and passed to the bifurcation of the main-stem bronchus. A 40-mL suspension of virulent EIDL 5–331 *R. equi* containing approximately 1 x $10^6$ viable bacteria was administered transendoscopically, with 20 ml infused into the right mainstem bronchus and 20 ml into the left mainstem bronchus via a sterilized silastic tube inserted into the endoscope channel. The silastic tube was flushed twice with 20 ml of air after each 20-ml bacterial infusion. Foals and their mares were housed individually in stalls and separately from other mare and foal pairs for 1 week following experimental infection. After 1 week, these mare/foal pairs were transferred back to their original pasture.

## Quantitative PCR

Trizol reagent (Invitrogen) was used for total RNA extraction according to the manufacturer's protocol. RNA was isolated using Direct-zol RNAeasy kits (Zymogen). cDNA was synthesized with BioRadiScript Direct Synthesis kits (BioRad) according to the manufacturer's protocol. qRT-PCR was performed in triplicate wells using PowerUp SYBR Green Master Mix. Data were analyzed on a QuantStudio 6 Real-Time PCR System (Applied Biosystems).

## RNA-sequencing

RNA was isolated from cells in biological triplicate using Trizol reagent (Invitrogen) and Direct-zol RNAeasy miniprep kit (Zymogen) according to the manufacturer's protocol. Agilent Technologies Bioanalyzer 2100 (Agilent Technology, Santa Clara, CA US) was used to verify RNA integrity number (RIN), rRNA ratio and RNA concentration. RNA-seq and library prep was performed by Texas A&M AgriLife Research Genomics and Bioinformatics Service. High-throughput RNA sequencing of samples was carried out on an Illumina NovaSeq6000S1

X using 2x 100-bp paired-end reads, which generated an average of ~32 million raw sequencing reads from each of 3 biological replicates for uninfected and infected macrophages. Analysis was performed as previously described[45]. Briefly, raw reads were filtered and trimmed and Fastq data was mapped to the Mus musculus Reference genome (RefSeq) using CLC Genomics Workbench (Qiagen). Differential expression analyses were performed using CLC Genomics Workbench. Relative transcript expression was calculated by counting reads per kilobase of exon model per million mapped reads (RPKM). Statistical significance was determined using the EDGE test via CLC Genomics Workbench. The differentially expressed genes were selected as those with a p-value threshold <0.05.

## ISRE reporter assay

Macrophage-secreted type I IFN levels were determined using a L929 cells stably expressing a luciferase reporter gene under the regulation of type I IFN signaling pathway (L929 ISRE cells). At the indicated times post-infection, macrophage cell culture media was harvested and stored at -80˚C. On the day prior to the assay, $5x10^4$ L929 ISRE cells were added to each well of a white 96-well flat-bottomed plate and incubated at 37˚C/5%CO$_2$ overnight. On the day of the bioassay, a 1:5 dilution of media from infected macrophages was added to each well of L929 ISRE cells, then incubated for 5h. Cells were washed with 1X PBS, lysed in reporter lysis buffer, then 30 μl of Luciferase Assay System solution (Promega) added and luminescence read immediately using a Cytation5 plate reader.

## IFN-β ELISA

Macrophage supernatant samples were harvested at 0-, 8-, and 24h post infection and stored at -80˚C until thawing on the day of the assay. Invitrogen Nunc MaxiSorp 96-well plates were coated with 50μl of capture antibody diluted 1:5000 (Santa Cruz, sc-57201) in 0.1M carbonate buffer and were incubated at 4˚C overnight. Wells were then blocked using PBS+10% FBS for 2h at 37˚C. Fifty μl of undiluted sample was added to each well. IFN-β standard (PBL, 12400–1) was diluted 1:4 for serial dilutions and incubated at room temperature (20˚C) overnight. Samples were washed with PBS+0.05% TWEEN before each step. Detection antibody (RnD Systems, 32400–1) was added at a 1:2000 dilution and incubated at room temperature (20˚C) overnight. After washing, secondary antibody (Cell signaling technology, 7074) was added to each well at a 1:2000 dilution and incubated for 3h. Following washing, the reaction was visually monitored until the standard was developed, then TMB substrate (SeraCare) was added and the reaction stopped with 2N H2SO4. The ELISA was read immediately at 450 nm using a BioTek plate reader.

## Immunoblot

Cell monolayers were washed with 1X PBS and lysed in 1X RIPA buffer (150 mM NaCl, 1.0% NP-40, 0.5% sodium deoxycholate, 0.1% SDS, 50 mM Tris, pH 8.0) with protease and phosphatase inhibitors (1 tablet per 10 ml; Pierce). DNA was degraded using 250 unitsbenzonase (EMD Milipore). Proteins were separated by SDS-PAGE and transferred to nitrocellulose membranes. Membranes were blocked for 1 h at RT in Odessey blocking buffer (Licor) or 4% BSA and incubated overnight at 4˚C with the following antibodies: pNF-κBSer536, 1:1000 (Cell Signaling 3033S), STAT1 1:1000 (Cell Signaling 9172S), pSTAT1 Tyr7011:1000 (Cell Signaling 9177S); IRF3 1:1000 (Cell Signaling 4302), pIRF3 Ser396 1:1000 (Cell Signaling 4947); Beta Actin 1:5000 (Abcam 6276), Tubulin 1:5000 (Abcam 179513). Membranes were washed 3x in 1X TBS 0.1% Tween 20 andincubated with appropriate secondary antibody (Licor) for 2 h at RT (20˚C) prior to imaging on Odessey Fc Dual-Mode Imaging System (Licor).

## Immunofluorescence microscopy

RAW 264.7 macrophages expressing epitope tagged galectins were seeded at $2x10^5$ cells/well on glass coverslips in 24-well dishes. At the indicated time point, cells were fixed in 4% para-formaldehyde for 10 min at 37˚C, then washed three times with PBS. Coverslips were incubated in primary antibody diluted in TBS+ 0.25% Triton-X + 5% Normal Goat Serum (TBST-NGS) for 3h. Cells were then washed three times in PBS and incubated in secondary antibodies diluted in TBST-NGS for 1h. Coverslips were incubated in DAPI for 5 minutes, then washed twice with PBS and mounted on glass slides using Fluoromount (Diagnostic Biosystems; K024) for imaging. Images were obtained using a FV1200 Olympus inverted confocal microscope equipped with 60X oil immersion objective. Images were analyzed using ImageJ. Maximum intensity projections of z-stacks were obtained and projected images were thresholded such that Flag-tagged galectin puncta in macrophages were masked and counted. To obtain percent colocalization, GFP-tagged *R. equi* 103 were thresholded until they were masked, and region of interest (ROI) was saved. The *R. equi* ROI was then applied to the thresholded galectin channel and measured. Results are expressed as percentage of bacteria colocalized with galectin.

## Statistics

All data are representative of at least 2 independent experiments with $n \geq 3$. Statistical analysis was performed using GraphPad Prism software (GraphPad, San Diego, CA). Two-tailed unpaired Student's t-tests were used for statistical analysis unless otherwise noted. Results are reported as the mean ± SD.

## Supporting information

**S1 Fig. *R. equi* induces pro-inflammatory cytokines in macrophages.** (A) Table of equine bronchoalveolar lavage (BAL) cell composition based on a 300-cell count differential. (B) RT-qPCR of *Actb* in murine BMDMs from experiment in Figs 1 and 2. (C) As in (B) but in RAW 264.7 cells. (D) As in (B) but in equine BAL cells. (E) As in (B) but *Myd88* KD RAW 264.7 cells. (F) RT-qPCR of *Tnfa* in *Myd88* KD RAW 264.7 cells stimulated or not with LPS for 4 hours. All RT-qPCRs are representative of at least 2 independent experiments and are the mean of 3 replicates ± SD, n = 3.
(TIF)

**S2 Fig.** (A) RT-qPCR of 16S in *R. equi* 33701+ and 33701- used as a housekeeping gene for Fig 3C. (B) RT-qPCR of *Actb* (housekeeping gene) in RAW 264.7 experiments in Fig 3. All RT-qPCRs are representative of at least 2 independent experiments and are the mean of 3 replicates. Error bars are ± SD, n = 3. Statistical significance was determined using Students' t-test. *p < 0.05, n.s. = not significant.
(TIF)

**S3 Fig. IFNAR is required for ISG expression in *R. equi*-infected murine macrophages.** (A) RT-qPCR of *Actb* in *R. equi*-infected TBK1 BMDMs. (B) As in (A) but *Tnfa* in IFNAR BMDMs. (C) As in (B) but *Ifnb* and *Isg15*. (D) RT-qPCR of *Isg15* in IFNAR BMDMs treated with ISD for 4h. (E) As in (B) but *Actb*. (F) RT-qPCR of *Trif* in RAW 264.7 KD cells relative to Scr control. (G) RT-qPCR of *Ifnb* and Isg15 in *Trif* KD and Scr RAW 264.7 macrophages. RT-qPCRs are representative of at least 2 independent experiments and are the mean of 3 replicates. Error bars are ± SD, n = 3. Statistical significance was determined using Students' t-test. *p < 0.05, n.s. = not significant.
(TIF)

**S4 Fig. The cytosolic DNA sensing axis of cGAS/STING/TBK1 is required to induce type I IFN in macrophages infected with *R. equi*.** (A) RT-qPCR of *Ifnb* in GFP gRNA control and STING KO RAW 264.7 macrophages treated or not with ISD for 4h. (B) ISRE reporter assay measuring relative luminescence units as a readout for type I IFN protein levels secreted into supernatants from GFP gRNA control or STING KO RAW 264.7 macrophages infected or not with *R. equi* for the indicated times or treated with ISD for 4h. (C) As in (A) but in GFP gRNA control and cGAS KO RAW 264.7 macrophages. (D) RT-qPCR of *Sting* in U937 cells relative to Scr control. (E) RT-qPCR of *Isg15* in STING U937 KD cells infected with R. equi for 8h. (F) As in (B) but in GFP gRNA control and cGAS KO RAW 264.7 macrophages. (G) CFUs of GFP gRNA control, cGAS KO and STING KO RAW 264.7 macrophages infected with *R. equi* at the indicated times. (H) RT-qPCR of *Actb* in GFP gRNA control, cGAS and STING KO RAW 264.7 cells at the indicated times post *R. equi* infection. (I) RT-qPCR of *Ifnb* and *Isg*15 in *R. equi*-infected RAW 264.7 cells depleted of mtDNA with 10 μM ddC for 4 days. (J) OD$^{600}$ of *R. equi* cultures treated with the indicated concentration of chloramphenicol. (K) RT-qPCR of *Ifnb* in RAW 264.7 cells 8h post infection with *R. equi* treated with the indicated concentration of chloramphenicol. (L) As in (K) but *Actb*. (M) As in (L) but RAW 264.7 cells treated with 100nM bafilomycin A1. (M) RT-qPCRs are representative of at least 2 independent experiments and are the mean of 3 replicates. Error bars are ± SD, n = 3. Statistical significance was determined using Students' t-test. $^*p < 0.05$, n.s. = not significant.
(TIF)

**S5 Fig.** (A) Immunofluorescence (IF) of RAW 264.7 cells stably expressing 3XFLAG (FL)-tagged galectin-3 or-8 infected with GFP expressing *R. equi* 103+ (MOI-5) for 16h.
(TIF)

**S1 Table. RNA-seq analysis of uninfected and *R. equi*-infected macrophages.** Positive and negative fold change are separated into tabs. Column A indicates the gene name. Column B indicates the *p*-value of the average of uninfected compared to *R. equi*-infected macrophages. Column C indicates the average fold change of uninfected vs infected macrophages. Columns D-F indicate individual fold change values for infected macrophages compared to the mean of uninfected macrophages.
(XLSX)

## Acknowledgments

We would like to thank members of the Patrick and Watson labs for their critical reading and feedback on this manuscript. We would like to acknowledge the Texas A&M AgriLife Research Genomics and Bioinformatics Service for performing our RNA-sequencing experiments. We gratefully acknowledge Dr. Mary Hondalus for provision of GFP expressing *R. equi* ATCC 103, and Kelsi West for *Trif* and *Myd88* KD macrophagecell lines.

## Author Contributions

**Conceptualization:** Krystal J. Vail, Noah D. Cohen, Angela I. Bordin, Kristin L. Patrick, Robert O. Watson.

**Data curation:** Krystal J. Vail.

**Formal analysis:** Krystal J. Vail.

**Funding acquisition:** Angela I. Bordin, Kristin L. Patrick, Robert O. Watson.

**Investigation:** Krystal J. Vail, Bibiana Petri da Silveira.

**Methodology:** Krystal J. Vail, Bibiana Petri da Silveira, Samantha L. Bell, Noah D. Cohen, Angela I. Bordin, Kristin L. Patrick, Robert O. Watson.

**Project administration:** Noah D. Cohen, Angela I. Bordin, Kristin L. Patrick, Robert O. Watson.

**Resources:** Samantha L. Bell, Angela I. Bordin, Robert O. Watson.

**Supervision:** Noah D. Cohen, Angela I. Bordin, Kristin L. Patrick, Robert O. Watson.

**Validation:** Krystal J. Vail, Samantha L. Bell.

**Visualization:** Krystal J. Vail, Robert O. Watson.

**Writing – original draft:** Krystal J. Vail.

**Writing – review & editing:** Krystal J. Vail, Bibiana Petri da Silveira, Samantha L. Bell, Noah D. Cohen, Angela I. Bordin, Kristin L. Patrick, Robert O. Watson.

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
