## [Decision Letter · Decision Letter 0]

7 May 2021

Dear %TITLE% Watson,

Thank you very much for submitting your manuscript "The opportunistic intracellular bacterial pathogen Rhodococcus equi elicits type I interferons by engaging cytosolic DNA sensing in macrophages" for consideration at PLOS Pathogens. As with all papers reviewed by the journal, your manuscript was reviewed by members of the editorial board and by several independent reviewers. In light of the reviews (below this email), we would like to invite the resubmission of a significantly-revised version that takes into account the reviewers' comments.

We cannot make any decision about publication until we have seen the revised manuscript and your response to the reviewers' comments. Your revised manuscript is also likely to be sent to reviewers for further evaluation.

Sincerely,

Thomas R. Hawn

Associate Editor

PLOS Pathogens

JoAnne Flynn

Section Editor

PLOS Pathogens

Kasturi Haldar

Editor-in-Chief

PLOS Pathogens

orcid.org/0000-0001-5065-158X

Michael Malim

Editor-in-Chief

PLOS Pathogens

orcid.org/0000-0002-7699-2064

Reviewer's Responses to Questions

**Part I - Summary**

Reviewer #1: The study by Vail et al, entitled “The opportunistic intracellular bacterial pathogen Rhodococcus equi elicits type I interferons by engaging cytosolic DNA sensing in macrophages” demonstrates that R. equi activates cytosolic DNA sensing in macrophages and elicits type I IFN responses in vitro and in vivo and it its potential impact in R. equi in vivo. This is an interesting study showing where similar results are observed in R. equi akin to what has been also described for M. tuberculosis. Said this, this study is important but there are some clarifications that authors must address and/or justify better.

Reviewer #2: The opportunistic intracellular bacterial pathogen Rhodococcus equi elicits type I interferons by engaging cytosolic DNA sensing in macrophages

Review on the work from Vail et al.,

In their study the authors investigated the mean by which the bacterial pathogen Rhodococcus equi is able to be detected and trigger a Type-I interferon pathway in its target cells, namely macrophages. The authors found that Rhodococcus equi-induced type I IFN-dependent ISG production requires cGAS/STING/TBK1 cytosolic pathway which suggests that this bacterium needs a direct or indirect access to the host cell cytosol. To this regard, the authors found that phagosomal permeabilization markers (Galectin 3, 8 and 9) were significantly recruited around several intracellular Rhodococcus equi, which suggests that through a yet to be determined mechanisms a small proportium of Rhodococcus equi has access to the host cell cytosol. In order to determine which virulence mechanisms/effectors might promote such response, the authors tested the well characterized virulence effector (VapA) encoded on the bacterium virulence plasmid, but did not find it required at modulating cGAS/STING/TBK1 activation path. Finally mouse and foal infections showed a robust ISG and IFN production pathway.

Overall, the study is well designed, executed and the findings that Rhodococcus equi also triggers a CGAS/STING path in a similar way to some Mycobacteria, the pathogen F.tularensis spp novicida etc… add a novel stone in the understanding of how microbial species interact to the intracellular environment.

Two important questions remain unfortunately not addressed in this study:

Is the activation of the cGAS/STING/IFN path important for the cell autonomous immunity of macrophages or for the general fight of the immune system against Rhodococcus equi?

Using mice or macrophages lacking cGAS or IFNaR or STING might help the authors to answer such question.

Another standing question from the study is if the putative ESX-encoded secretion system from Rhodococcus equi is able to trigger cGAS activation and phagosomal escape/destabilization? Is there a way the authors could try such approach (genetic against some putative compnents of this ESX?), as it would bring important scientific information regarding the general virulence components importants for driving such process

**Part II – Major Issues: Key Experiments Required for Acceptance**

Reviewer #1: 1/ For the RNA-seq study, the use of a mouse tumor cell line is understandable but removes the importance of using mouse primary alveolar macrophages to count for the lung environmental effects during infection, or perform an ex vivo analysis using isolated AMs after infection in mice. It seems that there is a disconnection between the RNA-seq results and the rest of the study. The discussion is not bringing together Fig 1 data with the rest. This study could easily start in Fig 2 defining the Th1/Th2/Th17 responses to R. equi during infection in vitro (or in vivo).

2/ Fig 2. Agree that there is a higher induction of these genes by 8 h (the latest time point measured) but it cannot be concluded that the peak of production was in this time point as later time-points were not studied. It will be also important to report the expression levels of their control (Actb) to fully understand how much is being produced, as well as report the data with the same y-axis graphs. If graphs in 2F had the same y-axis, will facilitate seeing that Liff1 is expressed higher in RAW264.7 cells and in primary BMDMs, and primary BMDMs have higher ISG expression that RAW264.7 (at least at 8 h) indicating that RNAseq data at 8 h in primary BMDMs may be a better read out for this analysis.

3/ Fig 3. It seems that although not significant (probably due to the n values reported and variability, large error bars), ifnb has a higher difference in expression at 4 and 8 h than il6 does, although the last one is significant. It will be important to have all graphs with the same y-axis scale for comparative reasons. Is there are R. equi overexpressing VapA? It is not clear the expression levels of VapA in 33701+. Data in Fig 2D and Fig. 3B for infb seems to be the same data set represented differently, if this is the case, authors could combine the results in the same graph to avoid confusions. Due to the minor differences in expression levels (although significant), to rule out that VapA is not being required for induction of Type I interferons, it may need to be necessary looking at protein and looking at both IFN-alpha and IFN-beta, as these are shown to be differentially expressed in these cells infected with the same virus.

4/ Fig S2. The presentation of this data set in the text may be misplaced and figure mislabeled. Further, completely abrogation is not observed for lsg15 in Fig. S2B, it is seeing a reduction related to scrabble treated cells. The same for ifnb levels, these are reduced at 4 h under one condition. Revise the discussion of this figure in the text. Revise data related to Supplemental S2A to make it more comprehensible to the general reader. Explain why Trif signaling is measured as a strong inducer of type I IFNs and regulator of gene expression of RANTES, IP-10, MCP-1 and IL-12p40 in response to the upstream TLRs.

5/ The cGAS/STING/TBK1 axis is already established for M. tuberculosis. Similarly, is there an interlink between the DNA (Esx-1 dependent) and RNA (SecA2/Esx-1 dependent) sensing pathways in stimulating Type I interferons as published for M. tuberculosis?

6/ Fig 6. There is observed bacterial clumping in these experiments. In the case of M. tuberculosis, clumping drives phagocytosis and phagosome-lysosome fusion, which can leak DNA/RNA and other cell wall Ags into the cytosol. It is difficult to be certain of the colocalization of R. equi with galactin-3 and 8 at 4 h due to the clumping. At 8 h, showed results are more definitive. The question is what this means, is R. equi being killed and their DNA/RNA translocated into the cytosol, is this only happening in the galactin + phagosomes (which represent a low number of events), and is this happening after lysosomal fusion, so in the phagolysosome containing R. equis?

7/ Fig 7, mouse study, it is not clear why responses were not measured in the lungs, where the infection normally establishes, as well as why Type I IFNs were not directly measured. The same applies to the foal model in Fig 8.

8/ As the authors mentioned there are still some important questions to answer, such as if it is in reality the R. equis DNA (or RNA, or both) involved in the stimulation of Type I IFNs or it is host DNA due to induced mitochondrial damage due to the infection.

Reviewer #2: Two important questions remain unfortunately not addressed in this study:

Is the activation of the cGAS/STING/IFN path important for the cell autonomous immunity of macrophages or for the general fight of the immune system against Rhodococcus equi?

Using mice or macrophages lacking cGAS or IFNaR or STING might help the authors to answer such question.

Another standing question from the study is if the putative ESX-encoded secretion system from Rhodococcus equi is able to trigger cGAS activation and phagosomal escape/destabilization? I sthere a way the authors could try such approach (genetic agsint some putative compnents of this ESX?), as it would bring great scientific information regarding the general virulence components importants for driving such process. However given the current context, if no mutant or this requires a very strong tour de force, an alternative way would be to determine using various inhibitors what in phagosomal lumen helps Rhodococcus equi to promote membrane damages. pH is important for F. novicida as well M. tuberculosis-activated cGAS, is such process also involved here? Does pH, ROS modulate phagosomal destabilization and cGAS activation?

**Part III – Minor Issues: Editorial and Data Presentation Modifications**

Reviewer #1: 9/ Line 242: This should be Fig. S3B.

10/ There are some issues identifying figures in the text, many are wrongly identified, specially supplemental Figure S2 and Fig 6C.

11/ Fig 6D, some pics are out of focus.

12/ Figure 4: Why did not study 8 h post-infection, where the peak in all the other results was reported? How many times this was done? This is not clear in the figure legend, although this info is in the Materials and Methods section.

13/ The strategy of using shRNA technology in mouse cell lines to KO for example MyD88 and not use primary cells from KO mice.

14/ Line 253-254: Revise if 'expression' should be 'productio'n and 'production' should be 'expression'.

15/ There is not need for Fig 7A and B, and Fig. 8A.

16/ Specifics about IAUC protocol numbers and foal housing is missing.

17/ Mouse infection study methodology is missing.

Reviewer #2: A minor point, but interesting, would be testing in primary human macrophages is Rhodococcus equi triggers cGAS-depndent IFN production and or cGAS-depndent macrophages necrosis as it has been shown that cGAS can mediate both through STING activation in primary human macrophages. https://www.cell.com/cell/pdf/S0092-8674(17)31133-9.pdf

PLOS authors have the option to publish the peer review history of their article (what does this mean?). If published, this will include your full peer review and any attached files.

Reviewer #1: No

Reviewer #2: No
---

## [Editor Report · Decision Letter 1]

12 Aug 2021

Dear %TITLE% Watson,

We are pleased to inform you that your manuscript 'The opportunistic intracellular bacterial pathogen Rhodococcus equi elicits type I interferons by engaging cytosolic DNA sensing in macrophages' has been provisionally accepted for publication in PLOS Pathogens.

Best regards,

Thomas R Hawn

Associate Editor

PLOS Pathogens

JoAnne Flynn

Section Editor

PLOS Pathogens

Kasturi Haldar

Editor-in-Chief

PLOS Pathogens

orcid.org/0000-0001-5065-158X

Michael Malim

Editor-in-Chief

PLOS Pathogens

orcid.org/0000-0002-7699-2064
---

## [Editor Report · Acceptance letter]

27 Aug 2021

Dear Watson,

We are delighted to inform you that your manuscript, "The opportunistic intracellular bacterial pathogen *Rhodococcus equi* elicits type I interferons by engaging cytosolic DNA sensing in macrophages," has been formally accepted for publication in PLOS Pathogens.

Best regards,

Kasturi Haldar

Editor-in-Chief

PLOS Pathogens

orcid.org/0000-0001-5065-158X

Michael Malim

Editor-in-Chief

PLOS Pathogens

orcid.org/0000-0002-7699-2064